# The peroxisome counteracts oxidative stresses by suppressing catalase import via Pex14 phosphorylation

Kanji Okumoto[1,2], Mahmoud El Shermely[1†], Masanao Natsui[2], Hidetaka Kosako[3], Ryuichi Natsuyama[2], Toshihiro Marutani[1], Yukio Fujiki[4,5]*

[1]Department of Biology, Faculty of Sciences, Kyushu University, Fukuoka, Japan; [2]Graduate School of Systems Life Sciences, Kyushu University, Fukuoka, Japan; [3]Division of Cell Signaling, Fujii Memorial Institute of Medical Sciences, Tokushima University, Tokushima, Japan; [4]Medical Institute of Bioregulation, Kyushu University, Fukuoka, Japan; [5]Institute of Rheological Functions of Food, Hisayama-machi, Fukuoka, Japan

*For correspondence:
yfujiki@kyudai.jp

Present address: †Basilea Pharmaceutica International Ltd, Basel, Switzerland

**Abstract** Most of peroxisomal matrix proteins including a hydrogen peroxide ($H_2O_2$)-decomposing enzyme, catalase, are imported in a peroxisome-targeting signal type-1 (PTS1)-dependent manner. However, little is known about regulation of the membrane-bound protein import machinery. Here, we report that Pex14, a central component of the protein translocation complex in peroxisomal membrane, is phosphorylated in response to oxidative stresses such as $H_2O_2$ in mammalian cells. The $H_2O_2$-induced phosphorylation of Pex14 at Ser232 suppresses peroxisomal import of catalase in vivo and selectively impairs in vitro the interaction of catalase with the Pex14-Pex5 complex. A phosphomimetic mutant Pex14-S232D elevates the level of cytosolic catalase, but not canonical PTS1-proteins, conferring higher cell resistance to $H_2O_2$. We thus suggest that the $H_2O_2$-induced phosphorylation of Pex14 spatiotemporally regulates peroxisomal import of catalase, functioning in counteracting action against oxidative stress by the increase of cytosolic catalase.

## Introduction

Peroxisome, an essential intracellular organelle, functions in various essential metabolism including β-oxidation of very long chain fatty acids and the synthesis of ether phospholipids (*Waterham et al., 2016*). Peroxisome contains a number of oxidases that generate hydrogen peroxide ($H_2O_2$) and catalase that decomposes $H_2O_2$ and potentially regulates reactive oxygen species (ROS) in the cell (*Schrader and Fahimi, 2006*). Peroxisomal functions rely on the tightly and spatiotemporally regulated import of the enzyme proteins responsible for respective reactions. Two topogenic signals are identified in the majority of peroxisomal matrix proteins: peroxisome targeting signal type-1 (PTS1) is a C-terminal tripeptide sequence SKL and its derivatives (*Gould et al., 1987*; *Miura et al., 1992*) and PTS2 is an N-terminal cleavable nonapeptide presequence (*Osumi et al., 1991*; *Swinkels et al., 1991*). Of 14 peroxisome assembly factors called peroxins in mammals, Pex14 is a peroxisomal membrane peroxin playing a central role in the import of both PTS1- and PTS2-proteins (reviewed in *Fujiki et al., 2014*; *Platta et al., 2016*). PTS1 receptor Pex5 recognizes newly synthesized PTS1-proteins in the cytosol. Pex14 acts as an initial target of the Pex5-PTS1-protein complex on peroxisomal membrane. By associating of Pex5 with the import machinery complexes in peroxisome membrane comprising Pex14, Pex13, and RING peroxins Pex2, Pex10 and Pex12, Pex5 transports its cargo proteins into the matrix, and then shuttles back to the cytosol (reviewed in *Fujiki et al., 2014*; *Liu et al., 2012*; *Platta et al., 2016*).

In mammals, catalase encoded by a single gene is a tetrameric heme-containing enzyme harboring an atypical PTS1, KANL, at the C-terminus (*Purdue and Lazarow, 1996*). Similar to typical PTS1 proteins, catalase is mainly localized to peroxisomes by the Pex5-mediated import pathway (*Otera and Fujiki, 2012*). Catalase forms fully active tetrameric conformation in the cytosol as noted in peroxisome-deficient fibroblasts (*Middelkoop et al., 1993*). Increased level of catalase is observed in the cytosol in aged human skin fibroblasts (*Legakis et al., 2002*). Furthermore, we recently reported that a proapoptotic protein BAK partially localizes to peroxisomes in mammalian cells and is involved in the release of catalase from peroxisomes (*Fujiki et al., 2017*; *Hosoi et al., 2017*). Although these findings suggest physiological importance of cytosolic catalase, molecular mechanisms underlying the regulation in translocation of peroxisomal matrix proteins remain largely unknown.

Posttranslational modification of protein regulates various functions of the cell in a fast, dynamic, and reversible fashion upon response to the changes in cellular demands and environmental conditions. Indeed, ubiquitination of Pex5 at a conserved cysteine residue is essential for its export from peroxisomes to the cytosol and peroxisomal matrix protein import (*Carvalho et al., 2007*; *Okumoto et al., 2011*). The cysteine residue at the position 11 of Pex5 is shown to be redox-sensitive, thereby Pex5-mediated PTS1 protein import can be regulated in the response to oxidative stress (*Apanasets et al., 2014*; *Walton et al., 2017*). As for another major posttranslational modification, that is phosphorylation, a large number of phosphorylation sites have been identified in various peroxisomal proteins by phosphoproteomic analysis in the yeast *Saccharomyces cerevisiae*, mouse, and humans (*Oeljeklaus et al., 2016*). Of these, phosphorylation of Pex14 is reported in the yeast *Hansenula polymorpha* (*Komori et al., 1999*; *Tanaka et al., 2013*) and *Pichia pastoris* (*Farré et al., 2008*; *Johnson et al., 2001*), but the biological importance and function remain unclear. In mammalian cells, mitogen-activated protein kinase (MAPK) pathways are shown to be activated in response to various oxidative stresses including ROS in the regulation of diverse cellular processes (*Ray et al., 2012*). Peroxisome is a $H_2O_2$-generating and -consuming organelle (*Fransen et al., 2012*), thus these findings suggest potential roles of ROS-dependent protein phosphorylation in regulating peroxisomal functions. It was reported that ataxia-telangiectasia mutated (ATM) kinase activated by ROS phosphorylates and subsequently ubiquitinates Pex5, thereby giving rise to degradation of peroxisomes, termed pexophagy (*Zhang et al., 2015*). However, how ROS plays a role in peroxisomal protein import remains undefined in any species.

Here, we address $H_2O_2$-dependent phosphorylation of mammalian Pex14. Phosphorylated Pex14 suppresses peroxisomal import of catalase, thereby functioning as an anti-oxidative stress response by elevating the level of catalase in the cytosol.

## Results

### Phosphorylation of Pex14 in mammalian cells

To investigate whether Pex14 is phosphorylated in mammalian cells, lysates of various mouse tissues were analyzed by electrophoresis using a conventional polyacrylamide gel (SDS-PAGE) and the one containing Phos-tag (hereafter described as Phos-tag PAGE). In Phos-tag PAGE, phosphorylated proteins can be distinguished as slower-migrating bands from the corresponding non-phosphorylated form (*Kinoshita et al., 2006*). We found that in Phos-tag PAGE, a Pex14 band with slower migration was readily discernible by immunoblotting in the lysates of mouse testis and liver (*Figure 1A*, upper panel, lanes 1 and 3, solid arrowhead) in addition to a similar level of unmodified Pex14 in both organs (open arrowhead). Pex14 was detected as a single band in conventional SDS-PAGE (*Figure 1A*, middle panel). The retarded-mobility form of Pex14 completely disappeared upon treatment with λ-protein phosphatase (*Figure 1A*, lanes 2 and 4), suggesting that Pex14 was partially phosphorylated in mammalian tissues, as observed in yeast (*Johnson et al., 2001*; *Komori et al., 1999*). Further, Phos-tag PAGE analysis showed that Pex14 in mouse tissues examined was phosphorylated at varying levels (*Figure 1B*), where the relatively higher phosphorylation was detected in liver and heart (*Figure 1B*, lanes 2 and 5). Similar phosphorylation profile of Pex14 was observed in rat hepatoma Fao cells under normal culture condition (*Figure 1C*, upper panel). Notably, we found that treatment with hydrogen peroxide ($H_2O_2$) increased the slower-migrating band of Pex14 with an additional lower mobility band in Phos-tag PAGE (*Figure 1C*, upper panel).

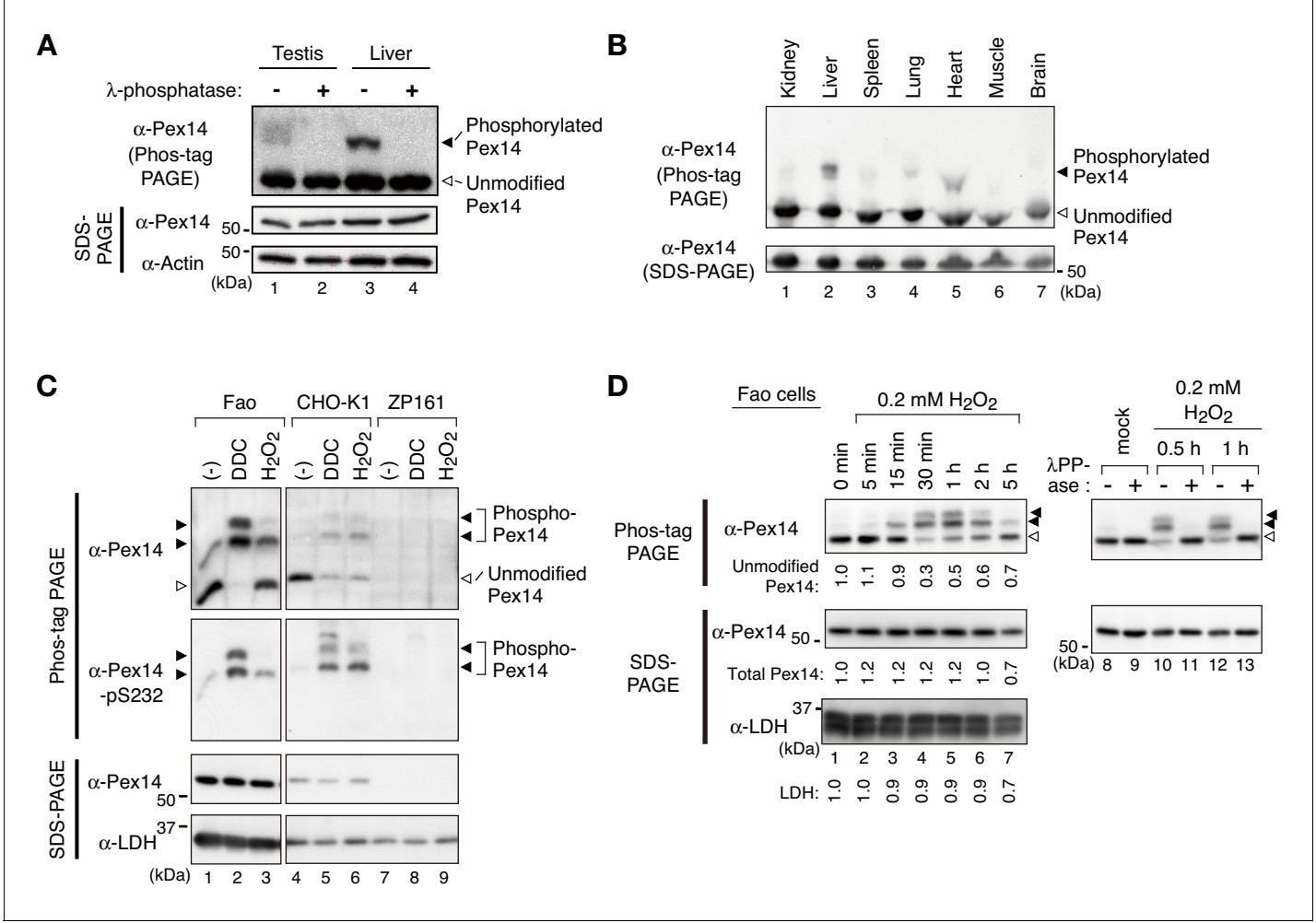

**Figure 1.** Pex14 is phosphorylated in vivo. (**A**) Lysates of testis and liver (20 µg each) from an 8-week-old male mouse were incubated with vehicle (-, lanes 1 and 3) and 400 unit λ-protein phosphatase (+, lanes 2 and 4). Samples were separated by Phos-tag PAGE (top panel) and SDS-PAGE (middle and bottom panels) and analyzed by immunoblotting with antibodies to Pex14 and actin, a loading control. Solid and open arrowheads indicate phosphorylated and unmodified Pex14, respectively. (**B**) Lysates of various mouse tissues (15 µg each) indicated at the top were analyzed by Phos-tag PAGE (upper panel), SDS-PAGE (lower panel), and immunoblotting with anti-Pex14 antibody. (**C**) Phosphorylation of Pex14 upon treatment with oxidative agents. Fao, CHO-K1, and a *PEX14*-deficient (*pex14*) CHO mutant ZP161 cells ($4 \times 10^5$ cells each) were treated for 30 min with vehicle (-), 100 µM diethyldithiocarbamate (DDC), and 1 mM hydrogen peroxide ($H_2O_2$). Cell lysates were analyzed as in A with antibodies to Pex14, phosphorylated Pex14 at Ser232 (Pex14-pS232), and lactate dehydrogenase (LDH). Open and solid arrowheads indicate unmodified and phosphorylated Pex14, respectively. Note that antibody to phsopho-Pex14 at Ser232 specifically recognized slower-migrating bands of Pex14 in Phos-tag PAGE. (**D**) *Left,* Time course of Pex14 phosphorylation upon $H_2O_2$ treatment. Fao cells were treated with 0.2 mM $H_2O_2$ as in B for indicated time periods. Unmodified Pex14 in Phos-tag PAGE and total Pex14 and LDH in SDS-PAGE were quantified and represented at the bottom of respective bands by taking as 1 those at 0 min. *Right,* λ-protein phosphatase treatment of phosphorylated Pex14. After the treatment with mock or 0.2 mM $H_2O_2$ for 0.5 hr and 1 hr, Fao cells were incubated with vehicle (-) and λ-phosphatase (+) as in A. The cell lysates were analyzed by Phos-tag PAGE (upper panels), SDS-PAGE (lower panels), and immunoblotting with anti-Pex14 antibody.

The online version of this article includes the following source data and figure supplement(s) for figure 1:

**Figure supplement 1.** Little effect of $H_2O_2$ treatment on Pex14 level and evaluation of Pex14 variants with anti-Pex14-pS232 antibody.

**Figure supplement 1—source data 1.** Data for graphs depicted in Figure supplement 1A.

Treatment of Fao and CHO-K1 cells with another oxidative agent, diethyldithiocarbamate (DDC), induced a nearly complete shift of Pex14 from the unmodified form to two slower-migrating bands (*Figure 1C*, upper panel). The $H_2O_2$- and DDC-dependent mobility shift of Pex14 in Phos-tag PAGE was likewise observed in CHO-K1 cells, to a similar extent between two oxidative agents (*Figure 1C*, upper panel). In contrast, as a negative control, neither slower-migrating bands nor unmodified

Pex14 were discernible in a *PEX14*-deficient (*pex14*) CHO mutant, ZP161 (*Shimizu et al., 1999*; *Figure 1C*, upper panel). ZP161 cells possess two types of deletions in the genome; a 133-base pair deletion in one allele created a termination codon at amino-acid residues 40–42 of Pex14, and an additional 103-base pair deletion in combination with the 133-base pair deletion in the other allele. Neither of two mutant forms of Pex14 is functional (*Shimizu et al., 1999*).

In Phos-tag PAGE using Fao cells, lower-migrating bands of Pex14 emerged at 15 min cell culture with 0.2 mM $H_2O_2$, peaked at 30 min to 1 hr, and gradually decreased to a nearly basal level within 5 hr (*Figure 1D*, left panels). In contrast to ~70% of total Pex14 was phosphorylated with a peak at 1 hr post-$H_2O_2$ challenge, total Pex14 level remained nearly constant in SDS-PAGE for 2 hr and reduced by approximately 30% at 5 hr after the $H_2O_2$ treatment, where a cytosolic protein LDH indicated a similar pattern (*Figure 1D*, left panels). We further verified any effect of $H_2O_2$ treatment on Pex14 stability and found that the exposure to $H_2O_2$ for 5 hr significantly lowered the protein level of Pex14 and concomitantly decreased LDH to a similar extent, thereby showing the relatively stable protein level of Pex14 (*Figure 1—figure supplement 1A*). The $H_2O_2$-induced Pex14 bands with lower mobility were sensitive to λ-protein phosphatase treatment and converged to the unmodified form (*Figure 1D*, right panels). These results strongly suggested that mammalian Pex14 is phosphorylated in vivo and that oxidative stresses such as $H_2O_2$-treatment transiently enhances the phosphorylation of Pex14. We further investigated the $H_2O_2$-stimulated phosphorylation of Pex14 and its functional consequence.

## Phosphorylation of Pex14 at Ser232, Ser247, and Ser252 is induced upon $H_2O_2$-treatment

Pex14 is a peroxisomal membrane protein containing a putative transmembrane segment and a coiled-coil domain (*Figure 2A*, upper diagram) (*Shimizu et al., 1999*; *Will et al., 1999*). Phos-tag PAGE analysis suggested that phosphorylation of Pex14 was highly induced at several sites upon $H_2O_2$ treatment of cells. It is known that various oxidative stresses activate MAPK signaling pathways (*Gaestel, 2006*; *Ray et al., 2012*). To identify the $H_2O_2$-inducible phosphorylation sites of Pex14, we expressed Pex14 mutants harboring alanine-substitutions for the potential phosphorylation residues that match with the consensus MAPK target sequence, Ser/Thr-Pro motif (*Gaestel, 2006*). When wild-type (WT) His-tagged rat Pex14 (His-Pex14) was expressed at a low level in Fao cells, $H_2O_2$-induced His-Pex14 phosphorylation was detected in Phos-tag PAGE (*Figure 2A*, lower panel), consistent with the case of endogenous Pex14 (*Figure 1*, C and D). In verifying various Ala-mutants of His-Pex14, a substitution of Ser232 to Ala (S232A) eliminated two slower-migrating bands of His-Pex14 (*Figure 2A*, lower panel). We raised an antibody that specifically recognized the phosphorylated Ser232 of Pex14 (*Figure 1—figure supplement 1B*) and demonstrated the $H_2O_2$-induced phosphorylation of endogenous Pex14 at Ser232 in Fao and CHO-K1 cells (*Figure 1C*). After $H_2O_2$-treatment of cells, His-Pex14-S232A still represented slower-migrating fuzzy bands in Phos-tag PAGE (*Figure 2A*, lower panel), suggesting that Ser232 is a major phosphorylation site of Pex14 and several other minor sites are present. To determine the $H_2O_2$-induced phosphorylation sites of endogenous Pex14, Pex14 was immunoprecipitated with anti-Pex14 antibody from vehicle- or $H_2O_2$-treated Fao cells, and the tryptic peptides were subjected to liquid chromatography-tandem mass spectrometry (LC-MS/MS) analysis. Four phosphorylated peptides corresponding to the amino acids at alignment positions 228–237 (QFPPpSPSAPK) (*Figure 2B*), 2–25 [Ap(SS)EQAEQPNQPSSSPGSENVVPR], 238–278 (IPSWQIPVKp(SPS)PSSPAAVNHHSSSDISPVSNESPSSSPGK), and 247–278 (SPSPp(SS)PAAVNHHSSSDISPVSNESPSSSPGK) (*Figure 2—figure supplement 1A*) were identified in $H_2O_2$-treated Fao cells, indicating the phosphorylation at Ser3 or Ser4, Ser232, Ser247 or Ser249, and Ser251 or Ser252 in Pex14. Label-free precursor ion quantification showed that Pex14 phosphorylation at Ser232, Ser247 or Ser249, and Ser251 or Ser252 increased by 21.7-, 104.7-, and 4.0-fold upon $H_2O_2$ treatment, respectively (*Figure 2C*), suggesting that Ser232, Ser247 or Ser249, and Ser251 or Ser252 were $H_2O_2$-induced phosphorylation sites of Pex14. These Ser residues except for Ser251 in rat Pex14 reside in a consensus Ser-Pro sequence of MAPKs target (*Gaestel, 2006*; *Figure 2D*). Ser232 of Pex14 is conserved in vertebrates, while Ser247 and Ser252 are relatively less conserved (*Figure 2D*). In contrast, $H_2O_2$ treatment gave rise to a small change in the phosphorylation of Pex14 at Ser3 or Ser4 (2.3-fold increase) (*Figure 2C*). $H_2O_2$-induced phosphorylation of Pex14 at Ser232 was also detected in several other cultured cell lines including human HepG2, HuH7, and HeLa cells, rat RCR1 cells, and mouse embryonic fibroblasts (MEF) (*Figure 2—figure*

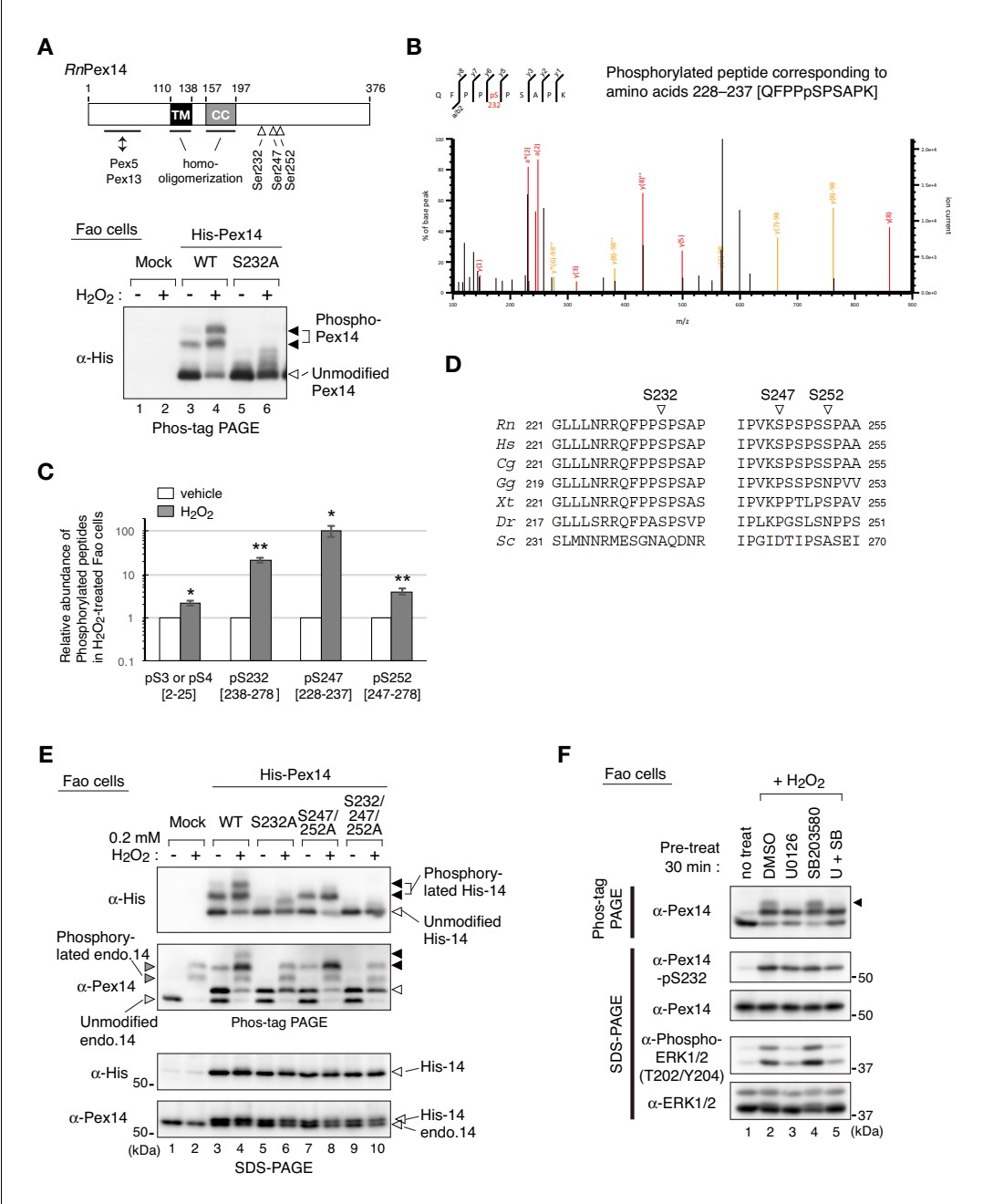

**Figure 2.** Hydrogen peroxide induces phosphorylation of Pex14 at three distinct serine residues in Fao cells. (A) *Upper*, a schematic view of domain structure of rat Pex14. Solid box, putative transmembrane (TM) domain; gray box, coiled-coil (CC) domain. *Lower*, Fao cells ($4 \times 10^5$ cells) were transiently transfected with plasmids encoding wild-type His-*Rn*Pex14 (WT) and S232A mutant harboring a substitution at Ser232 to Ala, and a mock plasmid (mock). At 24 hr after transfection, cells were treated for 30 min with vehicle (-) and 1 mM $H_2O_2$ (+) and the cell lysates were analyzed by Phos-tag PAGE and immunoblotting with anti-His antibody. Open and solid arrowheads indicate unmodified and phosphorylated forms of His-Pex14, respectively. (B) Mass spectrometric analysis of phosphorylated Pex14 induced by $H_2O_2$-treatment. Endogenous Pex14 in Fao cells ($8 \times 10^6$ cells) treated for 30 min with vehicle or 0.2 mM $H_2O_2$ was immunoprecipitated with anti-Pex14 antibody and subjected to LC-MS/MS analysis. Fragment spectrum of a phosphorylated peptide corresponding to amino acids 228–237 [QFPPpSPSAPK] showed phosphorylation of endogenous Pex14 at Ser232 (pS232) upon $H_2O_2$ treatment. (C) Quantification of phosphorylated Pex14 upon $H_2O_2$-treatment. Phosphorylated peptides were identified in Pex14 isolated from Fao cells that had been treated with vehicle or $H_2O_2$ as described in B. The levels of respective phosphopeptides in $H_2O_2$-treated cells (solid bars) were quantified with label-free precursor ion quantification and represented by taking as 1.0 that in vehicle-treated cells (open bars). Error bars represent means ± SEM of eight measurements in three independent experiments. *, p<0.05; **, p<0.01; unpaired Student's *t* test versus vehicle treated cells. (D) Multiple amino-acid sequence alignment of Pex14 neighboring Ser232, Ser247, and Ser252 of rat Pex14. (E) Fao cells transiently expressing wild-type His-Pex14 (WT) and the variants with indicated mutations were treated with 0.2 mM $H_2O_2$ for 30 min as in A and analyzed as in

*Figure 2 continued on next page*

*Figure 2 continued*

*Figure 1A* with antibodies to His and Pex14. Open and solid arrowheads were as in A. (**F**) Fao cells ($4 \times 10^5$ cells) pre-incubated for 30 min with vehicle (DMSO), 10 μM U0126, 10 μM SB203580, and 10 μM U0126 plus SB203580 were further treated with 0.2 mM $H_2O_2$ for 30 min. Cell lysates were analyzed as in *Figure 1A* by immunoblotting with indicated antibodies.

The online version of this article includes the following source data and figure supplement(s) for figure 2:

**Source data 1.** Data for the phosphorylated peptides of Pex14 shown in *Figure 2C*.

**Figure supplement 1.** $H_2O_2$-induced phosphorylation of Pex14: identification the phosphorylation sites by LC-MS/MS analysis and detection of Pex14-pS232 in various cultured cell lines.

**Figure supplement 2.** ERK2 siRNA shows no apparent effect on the $H_2O_2$-induced phosphorylation of Pex14.

*supplement 1B*), suggesting the highly conserved oxidative stress-inducible phosphorylation of Pex14 in mammals.

Next, three Ser residues, Ser232, Ser247, and Ser252, located in the C-terminal region of Pex14 (*Figure 2A*, upper diagram) were serially substituted to Ala to assess respective phosphorylation upon $H_2O_2$-treatment. His-Pex14-WT showed two phosphorylated bands in Phos-tag PAGE of $H_2O_2$-treated Fao cells and S232A mutation eliminated both bands (*Figure 2E*, lower panel of Phos-tag PAGE, lanes 4 and 6, solid arrowheads) as shown in *Figure 2A* (lower panel). In contrast, only the phosphorylated band with slower migration disappeared in the S247A/S252A double mutant (*Figure 2E*, lane 8), suggesting the phosphorylation at Ser247 and/or Ser257. Phosphorylated Pex14 was undetectable in the S232A/S247A/S252A triple mutant (*Figure 2E*, lane 10), consistent with the LC-MS/MS analysis (*Figure 2B*). Interestingly, the $H_2O_2$-induced phosphorylated band of endogenous Pex14 with slower migration was specifically eliminated by the pre-treatment with an ERK1/2 inhibitor U0126, but not with a p38 inhibitor SB203580 (*Figure 2F*, top panel, lanes 3 and 4, upper band (solid arrowhead)). Pre-treatment with both U0126 and SB203580 appeared to slightly reduce the level of phospho-Ser232 in Pex14 (*Figure 2F*, second upper panel). Transfection of *ERK2* siRNA to Fao cells reduced the protein level of ERK2 by ~80%, but showed no apparent effect on the phosphorylation level of Pex14 induced by $H_2O_2$ treatment (*Figure 2—figure supplement 2*, 5th panel). Upon $H_2O_2$ treatment, phosphorylation of ERK1 was instead induced in the ERK2-depleted cells at a level comparable to that of ERK2 in control siRNA-transfected cells, suggesting the complementation of ERK2 depletion by ERK1 (*Figure 2—figure supplement 2*, 4th panel). Taken together, these results suggest that Pex14 phosphorylation at Ser232 is primarily induced upon $H_2O_2$ treatment and that Ser247 and Ser252 are likely phosphorylated in an ERK-mediated manner.

## Phosphorylation of Pex14 suppresses peroxisomal import of catalase, not PTS1 proteins

We next investigated whether phosphorylation of Pex14 is involved in regulation of peroxisomal import of matrix proteins. In CHO-K1 and a CHO *pex14* mutant ZP161 (*Shimizu et al., 1999*), exogenous expression of Pex14 under a strong CMV promoter resulted in its phosphorylation without cell-treatment with $H_2O_2$ (*Figure 3—figure supplement 1A*). By introducing a modified CMV promoter lacking the enhancer region (*Okatsu et al., 2012*), we expressed Pex14 in ZP161 at a lower level including a phosphorylated Pex14 (*Figure 1—figure supplement 1B*; *Figure 3—figure supplement 1A*). With this weaker promoter, a series of Pex14 mutants with Ser-to-Ala (as phosphorylation-defective mutants) or Ser-to-Asp substitution (as phosphomimetic mutants) were transiently expressed in a *pex14* mutant ZP161. Immunoblot analysis showed that all Pex14 variants were expressed at a similar level as the wild-type Pex14 (*Figure 3A*). In immunofluorescence microscopy, Pex14-S232A mutant similarly restored peroxisomal import of catalase in ZP161 as wild-type Pex14, but Pex14-S232D mutant was lowered by about 50% in the restoring efficiency of the impaired catalase import (*Figure 3*, B and C). In contrast, both S232A and S232D mutations showed no significant difference in restoring of PTS1 protein import (*Figure 3C*; *Figure 3—figure supplement 1B*). Phosphorylation-deficient Pex14 mutants with either double mutation S247A/S252A or triple mutation S232A/S247A/S252A had no effect on respective restoring activity in catalase import, whereas the phosphomimetic triple mutant S232D/S247D/S252D, not the double mutant S247D/S252D, further lowered the efficacy than the single mutant S232D in the peroxisomal import of catalase (*Figure 3*, B and C). In the

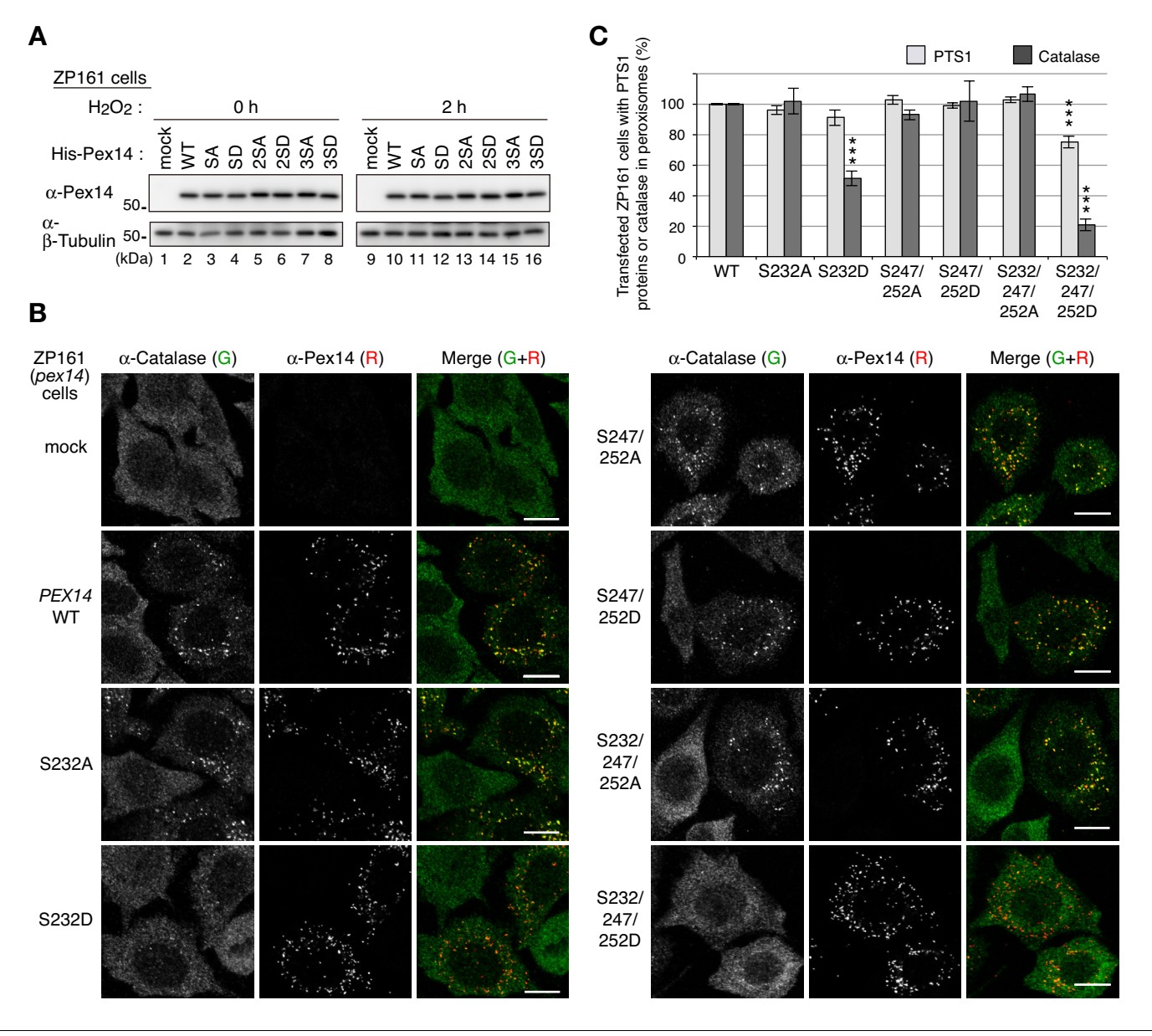

**Figure 3.** Phosphorylation of Pex14 suppresses peroxisomal import of catalase. (**A**) *pex14* ZP161 cells were transiently transfected for 36 hr with an empty vector (mock), wild-type (WT), and respective Ser mutants of *His-PEX14*. Cell lysates were analyzed by SDS-PAGE and immunoblotting with antibodies indicated on the left. (**B**) *pex14* ZP161 cells was transiently transfected with *PEX14* variants as in A. Cells were immunostained with antibodies to catalase (green) and Pex14 (red). Bar, 10 μm. (**C**) Quantification of the data in B and those for PTS1 proteins in *Figure 3—figure supplement 1B*. Percentages of the cells where PTS1 proteins (light gray) and catalase (dark gray) were mostly localized in peroxisomes in Pex14-expressing cells were represented as the means ± SEM by taking those as 100% in Pex14-WT-expressing cells. Transfected cells (n > 50) were counted in three independent experiments. ***p<0.001; one-way ANOVA with Dunnett's post hoc test versus cells expressing Pex14-WT.

The online version of this article includes the following source data and figure supplement(s) for figure 3:

**Source data 1.** Data for the import of catalase and PTS1 proteins shown in *Figure 3C*.

**Figure supplement 1.** Phosphomimetic Pex14-S232/247/252D mutation affects PTS1 protein import, but not peroxisomal localization of Pex14.

Pex14 mutants examined, peroxisomal import of PTS1 proteins was weakly,~20%, decreased in the triple mutant S232D/S247D/S252D (*Figure 3C*; *Figure 3—figure supplement 1B*). Co-immunostaining of Pex14 variants with a peroxisomal membrane protein PMP70 in ZP161 cells showed that all Pex14 variants were co-localized with PMP70-positive punctate structures,

peroxisomes, similarly to the wild-type Pex14 (*Figure 3—figure supplement 1C*). Together, these results suggested that Pex14 phosphorylation each at Ser232 and Ser247/Ser252 were mainly and additively involved in reducing peroxisomal import of catalase, respectively, with high specificity.

To further examine the effect of Pex14 phosphorylation on the regulation of catalase import, we established stable cell lines of *pex14* ZP161 each expressing wild-type His-Pex14 (named WT-6) and its mutants, phosphorylation-defective Pex14-S232A (SA-13) and phosphomimetic Pex14-S232D (SD-30) (*Figure 4A*). In these stable cell lines, wild-type Pex14 and the S232A and S232D mutants were expressed at similarly lower level (*Figure 4A*, top panel). A PTS1 protein, 75 kDa acyl-CoA oxidase (AOx) A-chain, is imported to peroxisomal matrix and proteolytically processed to 53 kDa B-chain and 22 kDa C-chain components (*Miyazawa et al., 1989*). AOx B-chain was discernible at an equal level in three Pex14 variant-expressing stable cell lines of ZP161 as in CHO-K1, but not detectable in *pex14* ZP161 (*Figure 4A*, upper middle panel) due to the instability of the A-chain in the cytosol (*Tsukamoto et al., 1990*), hence suggesting that Pex14-S232A and Pex14-S232D similarly restored peroxisomal import of AOx as the wild-type Pex14. Catalase expression level was indistinguishable between these three ZP161-stable cell lines, CHO-K1, and ZP161 (*Figure 4A*, lower middle panel). In immunofluorescence microscopy, catalase in CHO-K1 was localized in Pex14-positive punctate structures, peroxisomes, whereas in ZP161 catalase was detectable in the cytosol due to no expression of Pex14 (*Shimizu et al., 1999*; *Figure 4B*). As in CHO-K1 cells, catalase was predominantly detected in peroxisomes in stable cell lines, WT-6 and SA-13 expressing wild-type His-Pex14 and Pex14-S232A, respectively (*Figure 4B*). However, peroxisomal localization of catalase was severely lowered and cytosolic catalase was moderately elevated in the cell line SD-30 stably expressing Pex14-S232D (*Figure 4B*), where PTS1 proteins were almost exclusively detectable in peroxisomes as in CHO-K1, WT-6, and SA-13 cells (*Figure 4C*; *Figure 4—figure supplement 1A*), which was consistent with efficiently processed AOx as in CHO-K1 (*Figure 4A*). We further verified intracellular localization of peroxisomal matrix proteins by subcellular fractionation analysis. A higher level of catalase was detected in the cytosolic fraction (S) as compared to that in the organelle fraction (P) from the SD-30 cells expressing Pex14-S232D (*Figure 4D*, lanes 9 and 10, and *Figure 4E*). Consistent with our earlier report (*Hosoi et al., 2017*), a part of catalase was present in the cytosol fraction in CHO-K1 and in WT-6 cells expressing wild-type Pex14 (*Figure 4D*, lanes 1 and 5; *Figure 4E*). Catalase was barely detectable in the cytosolic fraction from SA-13 cells expressing Pex14-S232A (*Figure 4D*, lane 7). The ratio of cytosolic (S) to total (S + P) catalase of SA-13 cells indicated a significant decrease as compared to that of WT-6 cells (*Figure 4E*), suggesting that Pex14-S232A more efficiently imported catalase into peroxisomes than wild-type Pex14 that was partially phosphorylated in WT-6 cells (*Figure 1—figure supplement 1B*). Precursors of PTS2 proteins, peroxisomal fatty acyl-CoA thiolase and alkyldihydroxyacetonephosphate synthase (ADAPS), are converted to their respective mature forms in peroxisomes by cleavage of the amino-terminal PTS2 presequences (*de Vet et al., 1998*; *Honsho et al., 2008*; *Osumi et al., 1991*; *Swinkels et al., 1991*). Only mature forms of thiolase and ADAPS were detected at a similar level in the organelle fractions from CHO-K1 and three ZP161-stable cell lines (*Figure 4D*), demonstrating normal import of PTS2-proteins, similarly to the case of AOx import. These characteristics of peroxisomal matrix protein import in SD-30 cells were similarly observed in other two independent ZP161-stable cell lines expressing Pex14-S232D (data not shown). In immunofluorescence microscopy, Pex14 variants in three ZP161-stable cell lines including WT-6, SA-13 and SD-30 were normally localized to PMP70-positive peroxisomes (*Figure 4—figure supplement 1B*) as in the case of those transiently expressed in ZP161 (*Figure 3—figure supplement 1C*), showing no effect of these mutations in the peroxisomal localization of Pex14 variants. Collectively, these results suggested that phosphomimetic Pex14-S232D specifically reduces peroxisomal import of catalase, not PTS1- and PTS2-proteins.

To investigate the peroxisomal import of catalase upon $H_2O_2$ treatment in vivo, Fao cells were pulse-labeled with [35]S-labeled methionine and cysteine for 1 hr and fractionated into the cytosolic and organelle fractions. Immunoprecipitated [35]S-catalase was more detected in the cytosolic fraction than that in the organelle fraction at the start of chase (*Figure 4F*, lanes 1 and 2). At 1 hr chase in normal condition, the ratio of [35]S-catalase in the organelle fraction was increased (*Figure 4F*, lanes 2 and 4; *Figure 4G*; solid bar), indicating peroxisomal import of newly synthesized catalase. On the other hands, in the presence of $H_2O_2$, the ratio of [35]S-

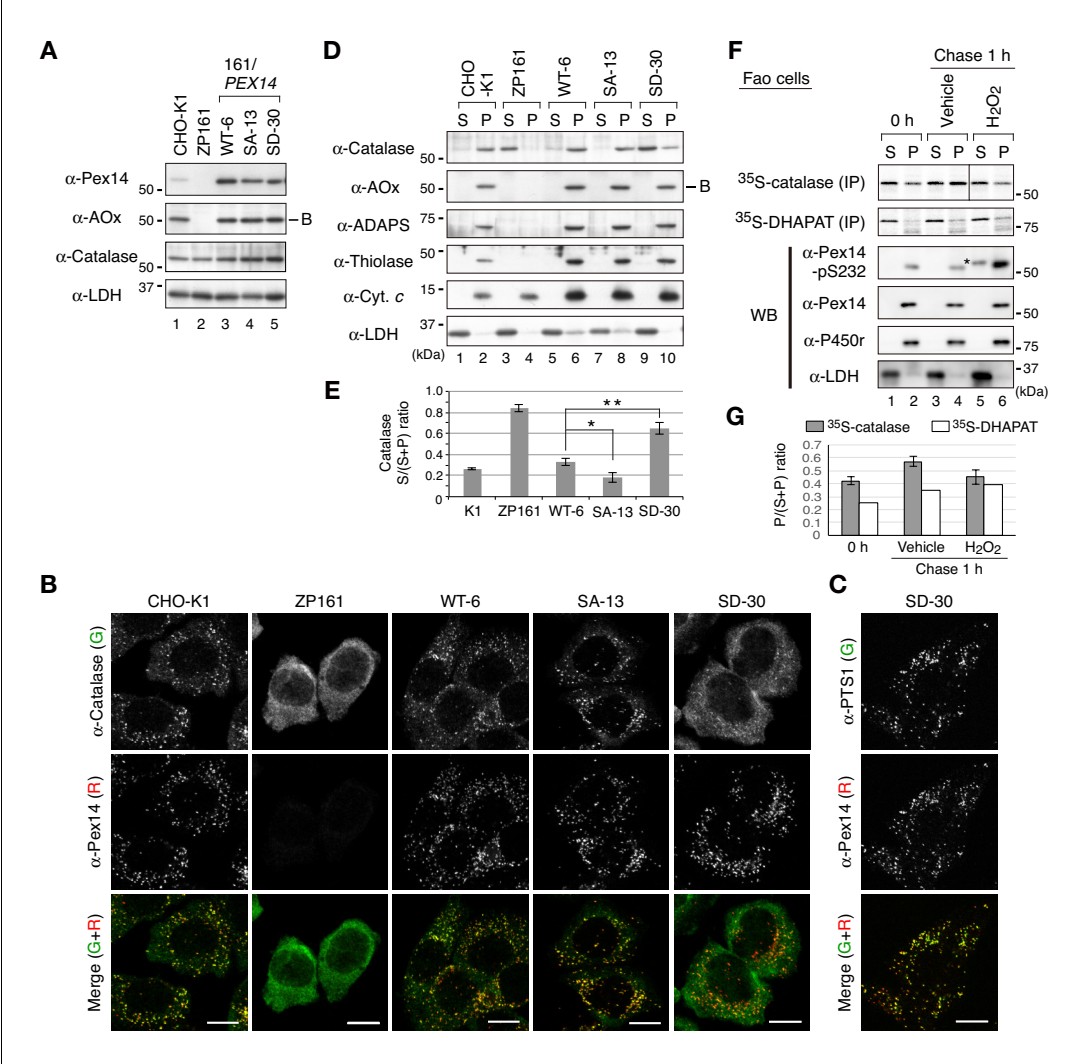

**Figure 4.** Phosphomimetic Pex14 mutant, Pex14-S232D, reduces catalase import into peroxisomes. (A) Cell lysates of CHO-K1, *pex14* ZP161, and stable cell lines of ZP161 each expressing wild-type His-Pex14 (WT-6) and its mutants, phosphorylation-deficient Pex14-S232A (SA-13) and phosphomimetic Pex14-S232D (SD-30) ($4 \times 10^5$ cells each) were analyzed by SDS-PAGE and immunoblotting with antibodies indicated on the left. Only the B-chain of acyl-CoA oxidase (AOx) that is generated by intraperoxisomal proteolytic processing of full-length AOx is shown. (B) Catalase was less imported into peroxisomes in Pex14-S232D-expressing cells. CHO-K1, *pex14* ZP161, and stable cell lines each expressing Pex14 variants were immunostained with antibodies to catalase (green) and Pex14 (red). Merged images were also shown. Bar, 10 μm. (C) Stable lines of *pex14* ZP161 expressing phosphomimetic Pex14-S232D (SD-30) were likewise immunostained with antibodies to PTS1 (green) and Pex14 (red). Bar, 10 μm. (D) Catalase in cytosolic fraction was increased in Pex14-S232D-expressing cells. Cells indicated at the top ($8 \times 10^5$ cells each) were separated into cytosolic (S) and organelle (P) fractions by permeabilization with 25 μg/mL digitonin and subsequent ultracentrifugation. Equal aliquots of respective fractions were analyzed by SDS-PAGE and immunoblotting with the indicated antibodies. AOx, a typical PTS1 protein; alkyl-dihydroxyacetonephosphate synthase (ADAPS) and 3-ketoacyl-CoA thiolase (thiolase), PTS2 proteins; Cyt. *c*, cytochrome *c*. LDH is a marker for cytosolic fraction. (E) Catalase level in the cytosolic and organelle fractions assessed in D was quantified and shown as a ratio of cytosol (S) to total (S plus P). Data represent means ± SEM of three independent experiments. Statistical analysis was performed by one-way ANOVA with Dunnett's post hoc test as compared with the S/(S+P) ratio of catalase in WT-6 cells. *p<0.05 and **p<0.01. (F) Pulse-chase experiment of catalase translocation. Fao cells were labeled with $^{35}$S-methionine and $^{35}$S-cysteine for 1 hr and were chased for 1 hr in the presence of vehicle or 0.2 mM $H_2O_2$. Cells were fractionated into the cytosol (S) and organelle (P) fractions as described in Material and methods. Equal aliquots of respective fractions were solubilized and subjected to immunoprecipitation with antibodies to catalase and DHAPAT. $^{35}$S-labeled catalase was analyzed by SDS-PAGE and detected by autoradiography (two upper panels). Equal aliquots of the cytosol and organelle fractions were analyzed by SDS-PAGE and immunoblotting using indicated antibodies. P450r, an ER membrane protein, cytochrome P450 reductase; LDH, a cytosolic protein. *, a putative nonspecific band. (G) $^{35}$S-labelled bands in F were quantified and $^{35}$S-catalase and $^{35}$S-DHAPAT in peroxisomes were shown as the ratio of respective bands in organelle (P) to total (S plus P). The P/(S+P) ratios of $^{35}$S-catalase and $^{35}$S-DHAPAT were represented as an average of two independent experiments and a single experiment, respectively.

The online version of this article includes the following source data and figure supplement(s) for figure 4:

*Figure 4 continued on next page*

Figure 4 continued

**Source data 1.** Data for the level of cytosolic catalase shown in Figure 4E and those of $^{35}$S-catalase and $^{35}$S-DHAPAT in the organelle fractions shown in Figure 4G.

**Figure supplement 1.** Phosphomimetic Pex14-S232D has no apparent effect on import of PTS1 protein and peroxisomal localization of Pex14.

**Figure supplement 2.** Pex5 recycling upon $H_2O_2$-treatment.

catalase in the organelle fraction was almost barely elevated for 1 h-chase (*Figure 4F*, lanes 5 and 6; *Figure 4G*; open bar), where Pex14 was phosphorylated (*Figure 4F*, lanes 5 and 6). By contrast, $^{35}$S-labeled dihydroxyacetonephosphate acyltransferase (DHAPAT), an enzyme with a typical PTS1, was increased in the organelle fractions during 1 hr chase in the absence or presence of $H_2O_2$ (*Figure 4*, F and G; open bar). These results strongly suggest that peroxisomal import of newly synthesized endogenous catalase is selectively suppressed by induction of Pex14 phosphorylation during the cell-exposure to $H_2O_2$.

Less efficient import of PTS1 proteins into peroxisomes and accumulation of Pex5 in peroxisome membrane are observed in human fibroblast cells at a late passage of cell culture. where intracellular ROS is elevated (*Legakis et al., 2002*). To assess the effect of $H_2O_2$ treatment on Pex5 recycling between peroxisomes and the cytosol, subcellular fractionation was performed under the condition that enabled to detect mono-ubiquitinated Pex5 at Cys11 (*Okumoto et al., 2011*). Pex5 in the organelle fraction was subtly and modestly increased upon 1 hr $H_2O_2$-treatment at concentration of 0.2 mM (standard setting otherwise mentioned in this study) and 0.5 mM, where both concentrations of $H_2O_2$ gave rise to phosphorylation of Pex14 at Ser232 to a similar extent (*Figure 4—figure supplement 2*, left panel, immunoblot with anti-Pex14-S232 antibody). Mono-ubiquitinated Pex5, which was detected as a DTT-sensitive slow-migrating band of Pex5 in the organelle fraction, was not altered in a ubiquitinated protein level upon $H_2O_2$ treatments (*Figure 4—figure supplement 2*, right panel, solid arrowhead). These results suggested that at least the treatment with 0.2 mM $H_2O_2$ for a short time period did not apparently affect the Pex5 recycling, as noted in the nearly normal import of $^{35}$S-PTS1 protein (*Figure 4F and G*). Collectively, upon treatment of cells with $H_2O_2$ the phosphorylation of Pex14 most likely suppresses peroxisomal import of catalase more selectively than that of PTS1 proteins.

## Phosphorylation of Pex14 selectively affects Pex5-mediated import complex formation with catalase

We next investigated the molecular mechanism underlying how phosphorylation of Pex14 regulates peroxisomal import of catalase. In mammals, N-terminal region of Pex14 is shown to interact with Pex5 and Pex13 (*Figure 2A*, upper diagram) (*Itoh and Fujiki, 2006*; *Otera et al., 2000*; *Schliebs et al., 1999*). Organelle fractions from Fao cells treated with vehicle or $H_2O_2$ were subjected to immunoprecipitation with anti-Pex14 antibody. Endogenous Pex14 was equally recovered from the cells with respective treatments, where the level of phosphorylated Pex14 at Ser232 was elevated in $H_2O_2$-treated cells (*Figure 5A*, two top panels, lanes 3 and 4). A higher level of Pex13 was included in the Pex14 complex from $H_2O_2$-treated cells than that from vehicle-treated cells, where Pex13 was expressed apparently at the same level (*Figure 5A*, lanes 1 and 2). Pex5 was increased in the organelle fraction upon $H_2O_2$ treatment (*Figure 5A*, lanes 1 and 2), but in the immunoprecipitates of Pex14 an equal but much low level of Pex5 was recovered between the prior to and upon the treatment with $H_2O_2$ (*Figure 5A*, lanes 3 and 4). In immunoprecipitation of Pex13, phosphorylated Pex14 was more efficiently recovered from $H_2O_2$-treated cells, while unmodified Pex14 was discernible at a similar level regardless of $H_2O_2$ treatment (*Figure 5A*, lanes 5–8; compare the recovery of phosphorylated Pex14 (solid arrowhead) to that of unmodified Pex14 (open arrowhead) in Phos-tag PAGE (top panel)). In immunoprecipitation with anti-Pex14-pS232 antibody, Pex13 was co-immunoprecipitated with phosphorylated Pex14, where a substantial amount of unmodified Pex14 was also associated with phosphorylated Pex14 (*Figure 5A*, lanes 9–12). These results suggested that $H_2O_2$-dependent phosphorylation of Pex14 increases its complex formation with Pex13 in the peroxisomal membrane. This is consistent with the finding that Pex13 is involved in peroxisomal import of catalase by interacting with Pex5 (*Otera and Fujiki, 2012*).

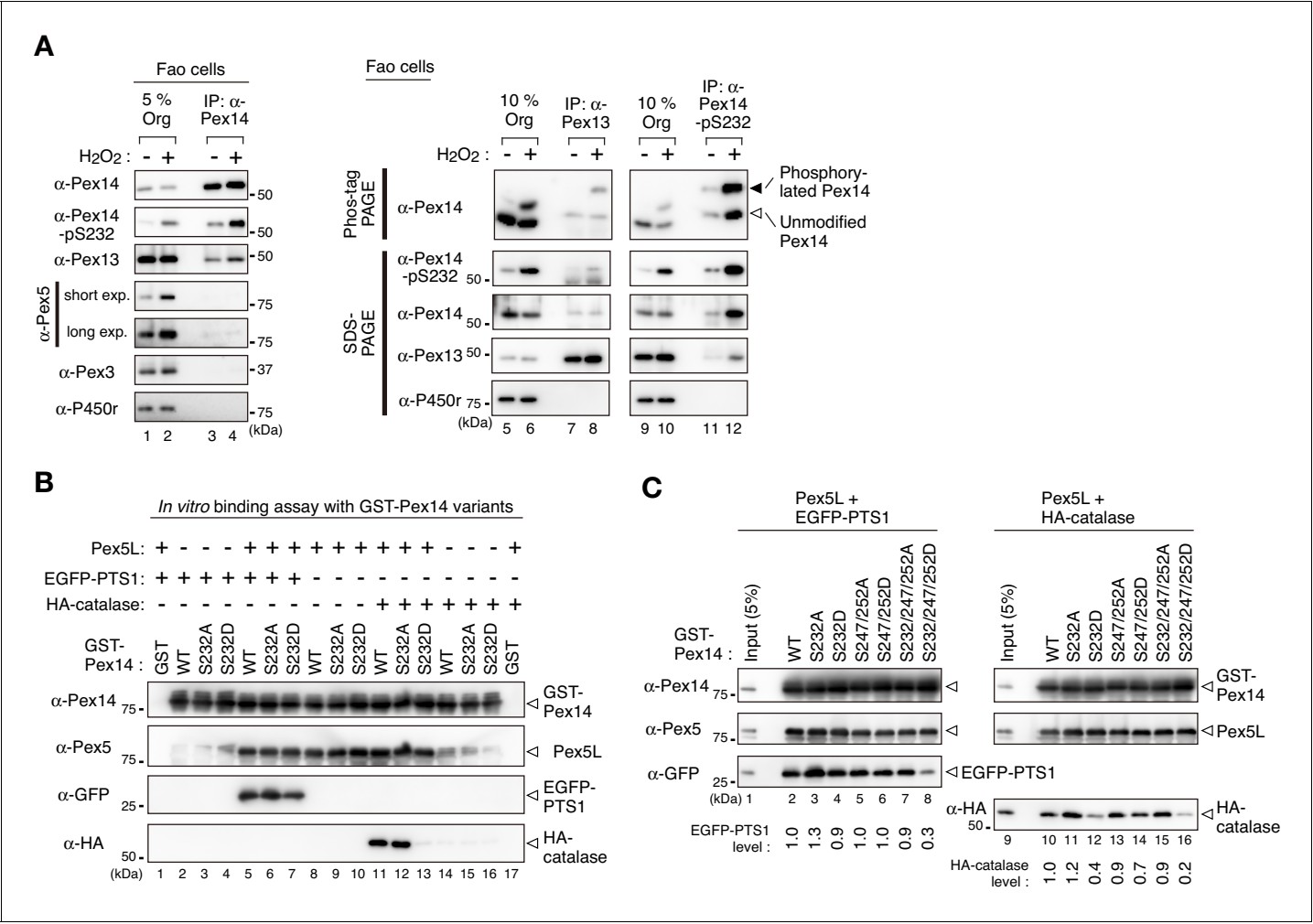

**Figure 5.** Phosphorylation of Pex14 exclusively affects Pex5-mediated complex formation with catalase. (A) Phosphorylated Pex14 forms a complex with Pex13. Organelle fractions of Fao cells ($4 \times 10^6$ cells each) treated for 30 min with vehicle (-) or 0.2 mM $H_2O_2$ were solubilized and subjected to immunoprecipitation with antibodies to Pex14 (lanes 3 and 4), Pex13 (lanes 7 and 8), and Pex14-pS232 (lanes 11 and 12). Equal-volume aliquots of immunoprecipitates (IP) and the input of organelle fractions (Org. input; 5% for IP of Pex14, 10% for IP of Pex13 and phosphorylated Pex14) were analyzed by SDS-PAGE and immunoblotting with antibodies indicated on the left. (B) In vitro binding assays were performed using recombinant proteins, that is GST-Pex14 variants, Pex5L, EGFP-PTS1, and HA-catalase. Components added to the assay mixtures, including GST in place of GST-Pex14 variants, are indicated at the top. Pex5L, EGFP-PTS1, and HA-catalase in the fractions bound to GST-Pex14-conjugated glutathione-Sepharose beads were analyzed by immunoblotting with antibodies indicated on the left. (C) In vitro binding assays were likewise performed using GST-Pex14 variants with mutations in three distinct Ser residues as in B. Five percent input of the reaction used was also loaded. Levels of the recovered EGFP-PTS1 and HA-catalase were quantified, normalized by that of GST-Pex14, and represented at the bottom by taking as one those pulled-down by GST-Pex14 WT.

The online version of this article includes the following figure supplement(s) for figure 5:

**Figure supplement 1.** S232D mutation in Pex14 affects Pex5-catalase interaction.

**Figure supplement 2.** An ATM inhibitor KU-55933 shows no effect on the $H_2O_2$-induced phosphorylation of Pex14.

Two isoforms of the PTS1 receptor Pex5, a shorter Pex5S and a longer Pex5L, function in mammals (*Otera et al., 1998*), both of which similarly recognize PTS1 cargo proteins including catalase and transports them to peroxisomes by docking on Pex14 (*Fujiki et al., 2014*; *Platta et al., 2016*). To further assess the effect of Pex14 phosphorylation in the interaction with known Pex14-binding partners, glutathione S-transferase (GST) pull-down assays were performed using recombinant proteins including Pex5 and the cargoes. In regard to Pex14-Pex5 interaction, Pex5L was equally detected in the fractions bound to GST-fused wild-type Pex14 (GST-Pex14-WT), GST-Pex14-S232A, and GST-Pex14-S232D, but not GST alone, suggesting that both S232A and S232D mutations have

no apparent effect on the direct binding of Pex14 to Pex5L (*Figure 5B*, lanes 1, 8–10, 17). As shown in the earlier report (*Otera and Fujiki, 2012*; *Otera et al., 2002*), PTS1 cargoes, both EGFP-PTS1 and HA-catalase, respectively formed a ternary complex with GST-Pex14-WT via Pex5L (*Figure 5B*, lanes 2, 5, 11, 14). However, GST-Pex14-S232D, not GST-Pex14-S232A, yielded an almost undetectable amount of HA-catalase in the bound fraction, despite the same-level recovery of Pex5L (*Figure 5B*, lanes 12 and 13). By contrast, in the presence of Pex5L, EGFP-PTS1 was detected in the fractions bound to GST-Pex14-S232D at a slightly lower level, as compared to that with GST-Pex14-WT and GST-Pex14-S232A (*Figure 5B*, lanes 5–7). In the absence of Pex5L, EGFP-PTS1 and HA-catalase were not detectable in the bound fractions of GST-Pex14 variants (*Figure 5B*, lanes 2–4, 14–16). Essentially the same results were obtained with a shorter isoform of Pex5, Pex5S (*Figure 5—figure supplement 1A,B*). Together, Pex14-S232D most likely interacts with Pex5 as wild-type Pex14 but it much less efficiently forms a Pex5-mediated ternary complex with catalase. We further investigated the effect of Pex14 phosphorylation at Ser247 and Ser252 on the ternary complex formation. None of the mutations S232A and S232D, double mutations S247A/S252A and S247D/S252D, and triple mutations S232A/S247A/S252A and S232D/S247D/S252D in Pex14 altered the binding efficiency to Pex5L (*Figure 5—figure supplement 1C*). HA-catalase was similarly pulled down with GST-Pex14 harboring respective mutations S247A/S252A and S232A/S247A/S252A in the presence of Pex5L, as seen with GST-Pex14-WT (*Figure 5C*, lanes 10, 13–15). On the other hands, HA-catalase bound to the Pex14-Pex5L complex was reduced to ~40% of wild-type Pex14 by S232D mutation alone and ~70% by S247/252D mutation (*Figure 5C*, lanes 10, 12, and 14). The triple mutant GST-Pex14-S232D/S247D/S252D showed further decrease of HA-catalase, to ~20% of wild-type Pex14, in the Pex5-dependent recovery (*Figure 5C*, lane 16). In forming the Pex5-mediated ternary complex with EGFP-PTS1, significant decrease was observed only with the GST-Pex14-S232D/S247D/S252D (*Figure 5C*, lanes 2–8). Nearly the same results were observed in the case with Pex5S (*Figure 5—figure supplement 1B*). Collectively, these results suggested that phosphorylation of Pex14 at Ser232 mainly suppresses the Pex5-mediated complex formation with catalase, where the phosphorylation at Ser247 and Ser252 likely provides additive effect. Moreover, the apparently simultaneous phosphorylation at the three sites also affect the complex formation with PTS1 proteins. Such distinct effects of Pex14 mutants on the ternary complex formation with catalase and typical PTS1 protein are in good agreement with the phenotypes of those observed in vivo, in cultured cells (*Figures 3* and *4*).

## Phosphorylation at Ser232 of Pex14 shows higher cell resistance to hydrogen peroxide

A part of catalase is localized in the cytosol even in the wild-type CHO-K1 cells, while catalase is fully diffused to the cytosol in peroxisome-defective mutants such as *pex14* ZP161 cells. Such cytosolic catalase is responsible for the cell resistance to exogenously added $H_2O_2$ (*Hosoi et al., 2017*). We next investigated whether $H_2O_2$-induced phosphorylation of Pex14 increases cytosolic catalase by suppressing its peroxisomal import in order to eliminate the cytosolic $H_2O_2$ for cell survival. At 16 hr after treatment of $H_2O_2$, cell viability in *pex14* ZP161 cells was higher than in CHO-K1 cells (*Figure 6*), as previously shown (*Hosoi et al., 2017*), and in ZP161-stable cell line WT-6 expressing wild-type Pex14

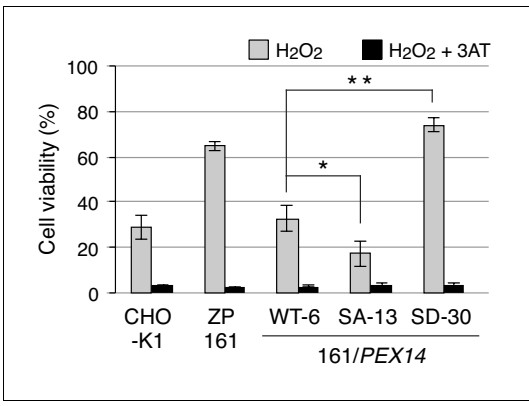

**Figure 6.** Phosphorylation at Ser232 of Pex14 is important for cell resistance to exogenous hydrogen peroxide. CHO-K1, *pex14* ZP161, and its stable cell lines expressing Pex14 variants ($1 \times 10^4$ cells each) were treated with 0.8 mM $H_2O_2$ in the absence (gray bars) and presence (solid bars) of 20 mM 3-aminotriazole (3AT), a catalase inhibitor. Cell viability was determined by MTS assay at 16 hr after $H_2O_2$ treatment and represented as percentages relative to that of each mock-treated, $H_2O_2$-untreated cells. Data represent means ± SEM of three independent experiments. *p<0.05 and **p<0.01; one-way ANOVA with Dunnett's post hoc test versus a stable cell line of ZP161 expressing Pex14-WT.

The online version of this article includes the following source data for figure 6:

**Source data 1.** Data for the cell viability upon $H_2O_2$ - treatment shown in Figure 6.

(*Figure 6*). A ZP161-stable cell line SD-30 expressing Pex14-S232D was more resistant to exogenous $H_2O_2$ like ZP161 than WT-6 cells, whereas a ZP161-stable cell line SA-13 expressing Pex14-S232A showed a significant decrease in the cell viability as compared with WT-6 cells (*Figure 6*). Moreover, these differential sensitivities to $H_2O_2$ between ZP161-stable cell lines were completely abrogated by the addition of 3-aminotriazole, a catalase inhibitor, indicative of catalase-dependent cell viability. The level of cell resistance to exogenous $H_2O_2$ (*Figure 6*) is well correlated with the amount of cytosolic catalase in respective types of cells (*Figure 4*, D and E). Taken together, these results suggest that oxidative stresses such as $H_2O_2$ enhance the phosphorylation of Pex14 at Ser232, thereby suppressing peroxisomal import of catalase and concomitantly elevating the cytosolic catalase to counteract $H_2O_2$ in the cytosol for cell survival.

## Discussion

Here our findings demonstrated the phosphorylation of mammalian Pex14 in response to $H_2O_2$ and assigned it as a novel regulation of peroxisomal import of catalase. We identified $H_2O_2$-induced three phosphorylation sites of Pex14, Ser232, Ser247, and Ser252 (*Figure 2B*; *Figure 2—figure supplement 1A*), all of which locate in the cytosolically faced, C-terminal region of Pex14 (*Figure 2A*, upper diagram). Our earlier domain mapping study suggested that the C-terminal region following the coiled-coil domain of Pex14 plays a role in peroxisomal protein import (*Itoh and Fujiki, 2006*). Notably, deletion of the residues at 201–367 including Ser232, not that at 261–367, abolished peroxisomal import of catalase but retained minimum import of PTS1 proteins (*Itoh and Fujiki, 2006*). These results support our view that C-terminal region of Pex14 has a regulatory role in peroxisomal protein import, which is modulated by the phophorylation of Ser232. Indeed, phosphorylation of Pex14 at Ser232 selectively lowers the Pex5-mediated complex formation with catalase (*Figure 5*, B and C). Pex14 forms a highly ordered homo-oligomer (*Itoh and Fujiki, 2006*) as well as large protein complexes of protein translocation machinery (*Meinecke et al., 2010*). Therefore, phosphorylation at Ser232 might induce the conformational change of Pex14-containing protein complexes, thereby leading to the predominant suppression of catalase import into peroxisomes. Phosphorylation at Ser247 and Ser252 potentially provides additive effect on the conformation of Pex14 complex, resulting in further inhibition in the peroxisomal import of catalase as well as partial retardation in that of PTS1 proteins (*Figure 3*, B and C; *Figure 3—figure supplement 1B*).

Upon $H_2O_2$ treatment, the level of Pex13 is increased in the immunoprecipitated Pex14 complexes containing highly phosphorylated Pex14 (*Figure 5A*). In mammals, Pex13 has been shown to play a pivotal role in peroxisomal import of catalase (*Otera et al., 2002*; *Meinecke et al., 2010*; *Otera and Fujiki, 2012*). A study using the Pex5 mutant defective only in the binding to Pex13 reported that Pex13-Pex5 interaction is specifically required for peroxisomal import of the folded and oligomeric proteins including catalase (*Otera and Fujiki, 2012*). A large complex comprising Pex5, Pex14, and Pex13 is suggested to form the pore-like structures in peroxisomal membrane to translocate a variety of peroxisomal matrix proteins (*Meinecke et al., 2010*). Together, these findings suggest that Pex13-Pex5 interaction is required for modulating the pore size to import the tetrameric catalase. In the present study, $H_2O_2$-induced phosphorylation of Pex14 is shown to incorporate a more amount of Pex13 into the Pex14-conating complexes as compared to those in normal condition (*Figure 5A*). Therefore, the phosphorylation of Pex14 might directly or indirectly modulates the Pex13-Pex5 interaction, suppressing the peroxisomal import of catalase. Although phosphorylation of Pex14 at its C-terminal part is identified in several yeast species, including Thr248 and Ser258 in *H. polymorpha* (*Komori et al., 1999*; *Tanaka et al., 2013*) and Ser266 and Ser313 in *S. cerevisiae* (*Albuquerque et al., 2008*), their functional roles remain to be defined. The Ser232 of Pex14 is conserved only in vertebrates (*Figure 2D*) and the amino-acid sequences of the C-terminal region of Pex14 share lower similarity between vertebrates and yeasts, hence suggesting that functional significance of the phosphorylation of Pex14 at C-terminal region is distinct between species. Yeast and worms have a cytosolic catalase in addition to a peroxisomal catalase harboring PTS1-like sequence (*Hartman et al., 2003*), implying that the modulated intracellular distribution of catalase is beneficial to the survival of organisms during the evolution.

ROS activates various cellular signaling pathways. ATM is reported as a peroxisome-localized kinase activated by ROS, mediating Pex5 phosphorylation and induction of pexophagy (*Zhang et al., 2015*). An ATM inhibitor KU55933 that abrogates $H_2O_2$-indeced phosphorylation of

Pex5 in HEK293 cells (*Zhang et al., 2015*) showed no effect on phopsphorylation of Pex14 upon the treatment with either $H_2O_2$ in Fao cells (*Figure 5—figure supplement 2*). Therefore, $H_2O_2$ most likely induces posttranslational modifications of Pex5 and Pex14 in a manner independent from ATM; the ATM-dependent phosphorylation and subsequent ubiquitination of Pex5 are distinct from the phosphorylation of Pex14 by the undefined kinase(s) reported here. Although the study with kinase inhibitors suggests the ERK-mediated phosphorylation of Pex14 at Ser247 and Ser252 (*Figure 2F*), further studies may be required to identify the kinase(s) that directly phosphorylates Pex14 in an $H_2O_2$-dependent manner and its upstream signaling pathway. Moreover, Pex14 phosphorylation upon cell treatment with $H_2O_2$ is transiently induced but is gradually reverted to the unmodified form (*Figure 1D*), where the turnover of Pex14 is not altered (*Figure 1D*; *Figure 1—figure supplement 1A*). This is further supported by the findings that phosphorylation-defective or phosphomimetic mutants of Pex14 were expressed at a similar level as the wild-type Pex14 even in $H_2O_2$-treated *pex14* ZP161 cells for 2 hr (*Figure 3A*, lanes 9–16). Collectively, phosphorylated Pex14 upon $H_2O_2$ treatment is most likely de-phosphorylated by yet identified phosphatase(s).

We show that $H_2O_2$-induced phosphorylation of Pex14 predominately suppresses peroxisomal import of catalase more efficiently than typical PTS1 proteins. Together with the findings that $H_2O_2$ treatment transiently induces Pex14 phosphorylation (*Figure 1D*) and that cell toxicity of $H_2O_2$ is more efficiently detoxified by cytosolic catalase (*Figure 6*; *Hosoi et al., 2017*), we propose a working model that Pex14 phosphorylation plays an essential role in the acute cell response against $H_2O_2$ challenge (*Figure 6*). This is possibly a regulatory system by taking advantage of the specific suppression of peroxisomal import of catalase, not that of PTS1 proteins, and the temporal increase of catalase in the cytosol by phosphorylation of Pex14. However, catalase chronically residing in the cytosol compromises the redox homeostasis in peroxisomes, thereby resulting in mitochondrial dysfunction or cell senescence (*Ivashchenko et al., 2011*; *Koepke et al., 2007*; *Walton et al., 2017*). Given the finding that catalase catabolizes $H_2O_2$ at the highest rate without effecting on the reducing equivalent such as glutathione (*Sies et al., 2017*), it is reasonable to have catalase function as the first defender in acute phase upon excess $H_2O_2$ insult. Oxidative stress such as $H_2O_2$ has been suggested to affect PTS1 protein import via Pex5 modification, depending on the redox state of the conserved Cys11 residue (*Apanasets et al., 2014*; *Walton et al., 2017*). Therefore, oxidative stress intrinsically lowers the import of PTS1 proteins and catalase under the imbalance of cellular redox. Collectively, the compromised import of catalase under the oxidative condition is most likely reflecting the severely impaired formation of ternary complex of Pex14, Pex5, and catalase (*Figure 5*, B and C) and a weaker affinity of Pex5 to catalase as compared to canonical PTS1 proteins (*Koepke et al., 2007*; *Otera and Fujiki, 2012*). In in vivo situation, both mechanisms might simultaneously take place for the cell survival, where suppression of peroxisomal catalase import and BAK-mediated release of catalase from the peroxisomal matrix are involved (*Hosoi et al., 2017*).

This report demonstrates that the protein import machinery of peroxisomes plays a crucial role in the regulatory network that counteracts the exogenous oxidative stresses. There are other intracellular sources of $H_2O_2$ involving mitochondria, NADPH oxidases, and peroxisomes (*Sies et al., 2017*). We observed that Pex14 phosphorylation at Ser232 was induced in Fao cells upon either treatments with rotenone, a mitochondrial complex I inhibitor, or peroxisome proliferators including clofibrate and bezafibrate that are known to elevate intracellular level of $H_2O_2$ (*Tada-Oikawa et al., 2003*; *Zhang et al., 2015*) (data not shown). Thus, $H_2O_2$ generated by various stimuli in response to the change of intracellular or extracellular environments could modulate and fine-tune the intracellular localization of catalase via Pex14 phosphorylation. In mitochondrial biogenesis, phosphorylation of yeast mitochondrial outer membrane proteins, Tom20, Tom22, and Tom70, indeed regulates the import of mitochondrial proteins in a nutrient condition-dependent manner (*Gerbeth et al., 2013*; *Schmidt et al., 2011*). Mice genetically overexpressing or deleting catalase reveal that catalase is also involved in various physiological and pathological processes such as renal injury (*Hwang et al., 2012*) and cardiomyocyte dysfunction (*Ye et al., 2004*) in diabetes. Together with the biological relevance of cytosolic catalase to mitochondrial dysfunction (*Ivashchenko et al., 2011*) and cell senescence (*Koepke et al., 2007*; *Walton et al., 2017*), a tackling issue needs to be addressed in regards to mechanisms underlying how Pex14 phosphorylation-dependent, spatiotemporal regulation and dysregulation of catalase are linked to the oxidative-stress state or age-related disease.

# Materials and methods

## Key resources table

| Reagent type (species) or resource | Designation | Source or reference | Identifiers | Additional information |
|---|---|---|---|---|
| Antibody | Rabbit polyclonal anti-FLAG | Sigma | F7425 RRID:AB_439687 | (1:1000) |
| Antibody | Rabbit polyclonal anti-phospho-Erk1/2 | Cell signaling | 9101S RRID:AB_331646 | (1:2000) |
| Antibody | Mouse monoclonal anti-Erk1/2 | Cell signaling | 4694S | (1:2000) |
| Antibody | Mouse monoclonal anti-FLAG (M2) | Sigma | F1804 RRID:AB_262044 | (1:1000) |
| Antibody | Mouse monoclonal anti-HA (16B12) | Covance | MMS-101R-200 RRID:AB_291263 | (1:2000) |
| Antibody | Mouse monoclonal anti-hexa-histidine tag | Qiagen | 34650 RRID:AB_2687898 | (1:500) |
| Antibody | Mouse monoclonal anti-GFP (B-2) | Santa Cruz Biotechnology | Sc-9996 | (1:1000) |
| Antibody | Mouse monoclonal anti-Tom20 (F-10) | Santa Cruz Biotechnology | sc-17764 RRID:AB_628381 | (1:1000) |
| Antibody | Mouse monoclonal anti-Cytochrome P450 reductase | Santa Cruz Biotechnology | sc-25270 RRID:AB_627391 | (1:2000) |
| Antibody | Mouse monoclonal anti-Cytochrome $c$ | BD Pharmingen | 556433 RRID:AB_396417 | (1:1000) |
| Antibody | Mouse monoclonal anti-ß-actin | MBL | M177-3 | (1:2000) |
| Antibody | Goat polyclonal anti-lactate dehydrogenase | Rockland | 110–1173 | (1:1000) |
| Antibody | Donley anti-Rabbit IgG, HRP-linked F(ab')$_2$ fragment | GE Healthcare | NA9340 RRID:AB_772191 | (1:4000) |
| Antibody | Sheep anti-Mouse IgG, HRP-linked whole Antibody | GE Healthcare | NA931 RRID:AB_772210 | (1:4000) |
| Antibody | Goat anti-Rabbit IgG (H+L) Secondary Antibody, Alexa Fluor 488 | Invitrogen | A11034 RRID:AB_2576217 | (1:10000) |
| Antibody | Goat anti-Guinea Pig IgG (H+L) Secondary Antibody, Alexa Fluor 568 | Invitrogen | A11075 RRID:AB_141954 | (1:10000) |
| Cell line (*C. griseus*) | CHO-K1 | *Tsukamoto et al., 1990* | | |
| Cell line (*C. griseus*) | *pex14* ZP161 | *Shimizu et al., 1999* | | A *PEX14*-deficient CHO mutant |
| Cell line (*C. griseus*) | ZP161 stably expressing His-*RnPEX14* WT (WT-6) | This paper | | A stable cell line of ZP161 expressing Pex14-WT |
| Cell line (*C. griseus*) | ZP161 stably expressing His-*RnPEX14* S232A (SA-13) | This paper | | A stable cell line of ZP161 expressing Pex14-S232A |
| Cell line (*C. griseus*) | ZP161 stably expressing His-*RnPEX14* S232D (SD-30) | This paper | | A stable cell line of ZP161 expressing Pex14-S232D |
| Cell line (*R. norvegicus*) | Fao | *Motojima et al., 1994* | | |
| Cell line (*R. norvegicus*) | RCR-1 | *Abe et al., 2020* | | |
| Cell line (*H. sapiens*) | HuH-7 | RIKEN | RCB1366 | |

*Continued on next page*

*Continued*

| Reagent type (species) or resource | Designation | Source or reference | Identifiers | Additional information |
|---|---|---|---|---|
| Cell line (*H. sapiens*) | HeLa | *Yagita et al., 2013* | | |
| Cell line (*H. sapiens*) | HepG2 | *Honsho et al., 2017* | | |
| Cell line (*M. musculus*) | MEF | *Itoyama et al., 2013* | | |
| Transfected construct (*R. norvegicus*) | siRNA to ERK2 | Sigma-Aldrich | SASI_Rn01_00107866 | GUAUAUACAUUCAGCUAAU |
| Recombinant DNA reagent | MISSION siRNA Universal Negative Control #1 | Sigma-Aldrich | SIC001 | |
| Recombinant DNA reagent | pCMVSPORT/*His-Rn PEX14 WT* (plasmid) | *Itoh and Fujiki, 2006* | | His-Pex14 WT |
| Recombinant DNA reagent | pCMVSPORT/*His-RnPEX14 SA or SD variants* (plasmid) | This paper | | His-Pex14 S232A, S232D, S247/252A, S247/252D, S232/247/252A, S232/247/252D |
| Recombinant DNA reagent | pcDNAZeo-D (plasmid) | This paper | | A mammalian expression vector with low transcription |
| Recombinant DNA reagent | pcDNAZeo-D/*His-Rn PEX14 WT* (plasmid) | This paper | | Wild-type His-Pex14 |
| Recombinant DNA reagent | pcDNAZeo-D/*His-RnPEX14 SA or SD variants* (plasmid) | This paper | | His-Pex14 S232A, S232D, S247/252A, S247/252D, S232/247/252A, S232/247/252D |
| Recombinant DNA reagent | pGEX/*RnPEX14 WT* (plasmid) | *Itoh and Fujiki, 2006* | | GST-Pex14 WT |
| Recombinant DNA reagent | pGEX/*His-RnPEX14 SA or SD variants* (plasmid) | This paper | | GST-Pex14 S232A, S232D, S247/252A, S247/252D, S232/247/252A, S232/247/252D |
| Recombinant DNA reagent | pGEX/*HA-Hs Catalase* (plasmid) | This paper | | for recombinant GST-HA-Catalase |
| Recombinant DNA reagent | pGEX/*ClPEX5S* (plasmid) | *Otera et al., 2002* | | |
| Recombinant DNA reagent | pGEX/*ClPEX5L* (plasmid) | *Otera et al., 2002* | | |
| Recombinant DNA reagent | pGEX/*EGFP-PTS1* (plasmid) | *Okumoto et al., 2011* | | |
| Recombinant DNA reagent | pGEX6P-1 (plasmid) | GE Healthcare | 28954648 | |
| Sequence-based reagent | Truncated CMV.Fw | This paper | PCR primer | ATGGGCGGTAGGCGTGTACG |
| Sequence-based reagent | Truncated CMV.Rv: | This paper | PCR primer | CGCGAAGCAGCGCAAAACG |
| Sequence-based reagent | RnPEX14-S232A.InvFw: | This paper | PCR primer | GCCCCGTCAGCCCCGAAGATCCCCTCCT- |
| Sequence-based reagent | RnPEX14-S232D.InvFw: | This paper | PCR primer | GACCCGTCAGCCCCGAAGATCCCCTCCT |
| Sequence-based reagent | RnPEX14-S232A/D.InvRv: | This paper | PCR primer | GGGAGGGAACTGTCTCCGATTC |
| Sequence-based reagent | RnPEX14-S252A.InvFw: | This paper | PCR primer | GCCCCCGCGGCCGTGAACCACCACAGC |
| Sequence-based reagent | RnPEX14-S252D.InvFw: | This paper | PCR primer | GACCCCGCGGCCGTGAACCACCACAGC |

*Continued on next page*

*Continued*

| Reagent type (species) or resource | Designation | Source or reference | Identifiers | Additional information |
|---|---|---|---|---|
| Sequence-based reagent | RnPEX14-S247_252A.InvRv: | This paper | PCR primer | GGAGGGTGACGGAGCCTTCACTGGG |
| Sequence-based reagent | RnPEX14-S247_252D.InvRv: | This paper | PCR primer | GGAGGGTGACGGGTCCTTCACTGGG |
| Sequence-based reagent | GST-HA-HsCatalase.BglFw: | This paper | PCR primer | GCGCAGATCTATGGCTTATCCATACGAC |
| Sequence-based reagent | pUcD3.Rv: | *Otera and Fujiki, 2012* | PCR primer | TTTCCACACCTGGTTGC |
| Chemical compound, drug | U0126 | Cell signaling | 9903S | |
| Chemical compound, drug | SB203580 | Cell signaling | 5633S | |
| Chemical compound, drug | KU-55933 | Abcam | ab120637 | |
| Chemical compound, drug | Compound C | Merck | 171260 | |
| Chemical compound, drug | Complete protease inhibitor cocktail | Roche | 11836170001 | |
| Chemical compound, drug | PhosStop phosphatase inhibitor cocktail | Sigma | 4906845001 | |
| Commercial assay or kit | CellTiter 96 AQueous One Solution Cell Proliferation Assay | Promega | G3580 | |
| Software, algorithm | R | R-project | http://www.r-project.org | |
| Software, algorithm | Image J | NIH | https://imagej.nih.gov/ij/ | |

## Cell culture, DNA transfection, and RNAi

CHO-K1 cell, a *pex14* CHO cell mutant ZP161 (*Shimizu et al., 1999*), rat astrocytoma RCR1 cell, and rat hepatoma Fao cell were cultured at 37°C in Ham's F-12 medium supplemented with 10% FBS under 5% $CO_2$ and 95% air (*Okumoto et al., 2011*). Human cervix epitheloid carcinoma HeLa, human hepatocellular carcinoma HepG2 and HuH7 cells, and MEF cells were cultured at 37°C in DMEM (Invitrogen) supplemented with 10% FBS (*Abe et al., 2018*). CHO and Fao cells were transfected with DNA using Lipofectamine reagent (Invitrogen) or polyethylenimine (PEI-MAX, Polysciences) according to the manufacturer's instructions. Stable transformants of ZP161 expressing rat Pex14 variants tagged with N-terminal hexahistidine (His-Pex14) were isolated by transfection of pcDNAZeo-D/*His-RnPEX14* variants (see below) followed by selection with Zeocin (Invitrogen), as described (*Okumoto et al., 2000*). Knockdown of ERK2 in Fao cells was performed by transfection of Mission siRNA (Sigma) with RNAiMax reagent according to the manufacturer's instruction. The sequence of siRNA for rat ERK2 is 5'-GUAUAUACAUUCAGCUAAU-3'. A siRNA Universal Negative Control #1 (Sigma) was used as a control. Fao cell (*Motojima et al., 1994*) was a kind gift from Dr. K. Motojima and HuH-7 cell was purchased from RIKEN BRC Cell Bank. CHO-K1 (*Tsukamoto et al., 1990*), ZP161 (*Shimizu et al., 1999*), RCR-1 (*Abe et al., 2020*), HeLa (*Yagita et al., 2013*), HepG2 (*Honsho et al., 2017*), and MEF (*Itoyama et al., 2013*) cells were as described. Cells were treated with a mycoplasma removal agent before experiments and were not subjected to mycoplasma testing.

## Plasmids

Plasmids encoding rat Pex14 variants with Ser-to-Ala or Ser-to Asp substitutions were generated by an inverse PCR method (*Weiner et al., 1994*) with KOD-plus DNA polymerase (Toyobo) and pCMVSPORT/*His-RnPEX14* (*Itoh and Fujiki, 2006*) as a template. To generate plasmids for weak Pex14 expression, upstream 564 bp of the CMV promoter was deleted from pcDNA3.1/Zeo

(Invitrogen) by an inverse PCR method, yielding a weaker expression vector, named pcDNAZeo-D. cDNAs encoding His-Pex14 variants in pCMVSPORT1 were ligated into the EcoRI-PstI sites in pcDNAZeo-D, generating pcDNAZeo-D vector-encoding His-Pex14 variants. Expression plasmids for GST-fused Pex14 variants were constructed by replacing the NheI-PstI fragment of wild-type *PEX14* in pGEX6P-2/*RnPEX14* (*Itoh and Fujiki, 2006*) with the corresponding fragment of *PEX14* variants in pCMVSPORT1 vector. To construct GST-fusion protein with catalase, the BglII–SalI fragment of *HA-Catalase* amplified by PCR from pUcD3/*HA-HsCatalase* (*Otera and Fujiki, 2012*) was cloned into the BamHI–SalI sites of pGEX6P-1 (GE Healthcare), thereby generating pGEX/*HA-HsCatalase*. Plasmids for GST-fusion proteins with Chinese hamster (*Cl*)Pex5S and *Cl*Pex5L (*Otera et al., 2002*) and EGFP-His-PTS1 (*Okumoto et al., 2011*) were also used. Primers used for PCR were shown in Key Resources Table.

## Antibodies and chemicals

Antibodies used were rabbit polyclonal antibodies each to C-terminal 19-amino acid residues of Pex14 (*Shimizu et al., 1999*), Pex5 (*Otera et al., 2000*), Pex13 (*Mukai and Fujiki, 2006*), Pex3 (*Ghaedi et al., 2000*), acyl-CoA oxidase (*Tsukamoto et al., 1990*), catalase (*Tsukamoto et al., 1990*), ADAPS (*Honsho et al., 2008*), DHAPAT (*Honsho et al., 2017*), 3-ketoacyl-CoA thiolase (*Tsukamoto et al., 1990*), PTS1 peptides (*Otera et al., 1998*), PMP70 (*Tsukamoto et al., 1990*), and guinea pig anti-Pex14 antibody (*Mukai et al., 2002*). Rabbit antiserum to phosphorylated Pex14 at Ser232, termed anti-Pex14-pS232 antibody, was raised in Biologica (Nagoya, Japan) by conventional subcutaneous injection of a synthetic 21-amino acid phosphopeptide comprising a 19-amino acid residues at 223–241 of rat Pex14 including a phospho-Ser232 and Gly-Cys di-peptide sequence at the C-terminus that had been linked to keyhole limpet hemocyanin (*Tsukamoto et al., 1990*). The raised rabbit antibody was purified in Biologica by affinity chromatography using a column conjugated to the synthetic phosphopeptide antigen after passing thorough that conjugated to the corresponding unmodified peptide. We purchased rabbit polyclonal antibodies to FLAG (Sigma-Aldrich) and phospho-Erk1/2 (Cell Signaling), mouse monoclonal antibodies to HA (16B12; Covance), FLAG (Sigma-Aldrich), hexa-histidine tag (Qiagen), GFP (Santa Cruz Biotechnology, Inc), Erk1/2 (Cell Signaling), cytochrome P450 reductase (Santa Cruz Biotechnology, Inc), cytochrome *c* (BD Pharmingen), β-actin (MBL), and Tom20 (Santa Cruz Biotechnology, Inc), and goat anti-lactate dehydrogenase antibody (Rockland). Kinase inhibitors, U0126 and SB203580, were purchased from Cell Signaling. KU5933 and Compound C were from Abcam and Merck, respectively.

## Preparation of mouse tissues and Phos-tag PAGE

Several different tissues from an 8 week old male mouse that had been fed with normal chow under regular day-light and dark cycle were directly lysed in buffer-L (20 mM HEPES-KOH, pH 7.4, 0.15 M NaCl, 25 µg/ml each of leupeptin and antipain, 1 mM phenylmethylsulfonyl fluoride (PMSF), and 1 mM dithiothreitol) containing 0.5% Nonidet P-40% and 0.1% SDS by ten strokes of homogenization with an Elvehjem-Potter homogenizer (*Miura et al., 1992*). After centrifugation, solubilized fractions (15 µg) were subjected to SDS-PAGE as described (*Natsuyama et al., 2013*). In phosphatase treatment, the soluble fractions were incubated with 1 µg/ml of λ-protein phosphatase (New England Biolab) for 30 min at 30 ˚C. Phos-tag PAGE was performed with 7.5% polyacrylamide gels containing 50 µM Phos-tag (Wako Chemicals) and 100 µM $MnCl_2$ for the lysates of mouse tissues and 25 µM Phos-tag and 50 µM $MnCl_2$ for those of cultured cells (*Kinoshita et al., 2006*).

## Mass spectrometry analysis

Fao cells ($8 \times 10^6$ cells) were lysed in RIPA buffer (50 mM Tris-HCl, pH7.6, 0.15 M NaCl, 1% Nonidet P-40, 0.5% sodium deoxycholate, 0.1% SDS, and 1 mM DTT) supplemented with a complete protease inhibitor cocktail (Roche) and Phos-stop phosphatase inhibitor cocktail (Sigma-Aldrich) and were flash-frozen in liquid nitrogen. After thawing on ice and centrifugation at 20,000 *g* for 15 min at 4 ˚C, supernatant fractions were subjected to immunoprecipitation with anti-Pex14 antibody immobilized on SureBeads Protein G magnetic beads (Bio-Rad) for 3 hr at 4 ˚C with rotation. After washing with RIPA buffer four times and then with 50 mM ammonium bicarbonate twice, proteins on the beads were digested by adding 400 ng trypsin/Lys-C mix (Promega) for 16 hr at 37 ˚C. The digests were

acidified and desalted using GL-Tip SDB (GL Sciences). The eluates were evaporated in a SpeedVac concentrator and dissolved in 3% acetonitrile (ACN) and 0.1% trifluoroacetic acid.

LC-MS/MS analysis of the resultant peptides was performed on an EASY-nLC 1200 UHPLC connected to a Q Exactive Plus mass spectrometer equipped with a nanoelectrospray ion source (Thermo Fisher Scientific). The peptides were separated on a 75 µm inner diameter x 150 mm C18 reversed-phase column (Nikkyo Technos) with a linear 4–28% ACN gradient for 0–100 min followed by an increase to 80% ACN for 10 min. The mass spectrometer was operated in a data-dependent acquisition mode with a top 10 MS/MS method. MS1 spectra were measured with a resolution of 70,000, an automatic gain control (AGC) target of $1 \times 10^6$ and a mass range from 350 to 1,500 $m/z$. HCD MS/MS spectra were acquired at a resolution of 17,500, an AGC target of $5 \times 10^4$, an isolation window of 2.0 $m/z$, a maximum injection time of 60 ms and a normalized collision energy of 27. Dynamic exclusion was set to 10 s. Raw data were directly analyzed against the SwissProt database restricted to *H. sapiens* using Proteome Discoverer version 2.3 (Thermo Fisher Scientific) with Mascot search engine version 2.5 (Matrix Science) for identification and label-free precursor ion quantification. The search parameters were as follows: (i) trypsin as an enzyme with up to two missed cleavages; (ii) precursor mass tolerance of 10 ppm; (iii) fragment mass tolerance of 0.02 Da; (iv) carbamidomethylation of cysteine as a fixed modification; and (v) acetylation of the protein N-terminus, oxidation of methionine and phosphorylation of serine, threonine, and tyrosine as variable modifications. Peptides were filtered at a false-discovery rate of 1% using the percolator node. Normalization was performed such that the total sum of abundance values for each sample over all peptides was the same.

## Immunofluorescence microscopy

Immunostaining of cells was performed as described (*Okumoto et al., 2011*) with 4% paraformaldehyde for cell fixation and 0.1% Triton X-100 for permeabilization. Immuno-complexes were visualized with an Alexa Fluor 488-labeled goat anti-rabbit IgG antibody and an Alexa Fluor 568-labeled goat anti-guinea pig IgG antibody (Invitrogen). Cells were observed by a confocal laser microscope (LSM710 with Axio Observer Z1; Zeiss) equipped with a Plan Apochromat 100 × 1.4 NA oil immersion objective lens and argon plus dual HeNe lasers at RT. Images were acquired with Zen software (Zeiss) and prepared using Photoshop (CS4; Adobe).

## Subcellular fractionation and immunoprecipitation

For separation of cytosolic and organelle fractions from CHO cells, harvested cells were incubated with 25 µg/ml digitonin in buffer H (20 mM Hepes-KOH, pH 7.4, 0.25 M sucrose, 1 mM DTT, complete protease inhibitor cocktail [Roche], 1 mM NaF, 1 mM $Na_3VO_4$, and 6 mM β-glycerophosphate) for 5 min at room temperature as described (*Natsuyama et al., 2013*). After centrifugation at 20,000 *g* for 30 min at 4 ℃, equal aliquots of respective fractions were analyzed by immunoblotting. For isolation of organelle fraction of Fao cells,~$4\times10^6$ cells were homogenized with a Potter-Elvehjem Teflon homogenizer (Wheaton) in buffer H and centrifuged at 800 *g* for 10 min at 4℃ to yield post nuclear supernatant (PNS) fraction. Organelle fraction was separated by ultracentrifugation of the PNS fraction at 100,000 *g* for 30 min at 4℃. The organelle pellet was lysed in buffer L containing 0.5% CHAPS, 1 mM NaF, 1 mM $Na_3VO_4$, and 6 mM β-glycerophosphate for 30 min at 4℃. After centrifugation at 20,000 *g* for 10 min at 4℃, resulting supernatants were incubated with antibodies to Pex14, Pex14-pS232, and Pex13 in buffer L containing 0.5% CHAPS for 2 hr at 4℃. Antibody-antigen complexes were recovered by incubating for 1 hr at 4℃ with Protein A-Sepharose CL-4B (GE Healthcare) and eluted with Laemmli sample buffer. For detection of mono-ubiquitinated Pex5 in organelle fraction of Fao cells, subcellular fractionation was performed with H buffer containing 5 mM N-ethylmaleimide and no DTT as described (*Okumoto et al., 2011*).

## In vitro binding assay

Pex14, Pex5S, Pex5L, EGFP-PTS1, and HA-catalase were expressed as GST fusion proteins in *Escherichia coli* DH5α and were purified with glutathione-Sepharose beads (GE Healthcare), as described (*Otera et al., 2002*). Pex5S, Pex5L, EGFP-PTS1, and HA-catalase were isolated from the purified GST fusion proteins by cleaving with PreScission protease (GE Healthcare) according to the manufacturer's protocol. GST or GST-Pex14 variants (typically 2 µg each) conjugated to glutathione-

Sepharose beads were incubated with Pex5 (2 µg), Pex13 (0.1 µg), EGFP-PTS1 (4 µg), or HA-catalase (4 µg) by rotating for 2 hr at 4°C in an in vitro binding buffer (50 mM Tris-HCl, pH 7.5, 0.15 M NaCl, 1% Triton X-100, 10% glycerol, 1 mM PMSF, 1 mM EDTA, and 1 mM DTT). Glutathione-Sepharose beads were washed three times with the binding assay buffer minus glycerol and the bound fractions were eluted with Laemmli sample buffer.

### Cell viability assay

Cell viability was measured with a tetrazolium-based toxicology assay kit (Promega). CHO cells ($1 \times 10^4$ cells per well) were seeded in a 96-well plate and grown for 24 hr and then treated for 14 hr with vehicle alone, 0.8 mM $H_2O_2$ alone, or each together with 20 mM 3-aminotriazole (3-AT). After treatment, cells were incubated for additional 2 hr with Celltiter 96 aqueous one solution reagent (Promega). Cell viability was determined by the absorbance of media at 490 nm as described in the manufacturer's protocol and represented as percentages relative to that of each mock-treated and $H_2O_2$-untreated cells.

### Pulse-chase experiment

Fao cells growing in DMEM supplemented with 10% FBS in 6-well plate were washed twice with PBS, incubated in cysteine- and methionine-free DMEM (Gibco) supplemented with 10% FBS that had been dialyzed for 1 hr in excess PBS with a Slide-A-Lyzer dialysis cassette (Thermo Fisher Scientific). Cells were then pulse-labeled for 1 hr by adding 100 µCi/ml $^{35}$S-methionine plus $^{35}$S-cysteine (American Radiolabeled Chemicals). To chase the $^{35}$S-labeled proteins, cells were washed twice and further incubated for 1 hr with DMEM supplemented with 10% FBS and 10 mM methionine. $^{35}$S-labeled cells were harvested and incubated for 5 min in buffer H containing 50 µg/ml digitonin at room temperature as described (*Natsuyama et al., 2013*). After centrifugation at 20,000 *g* for 30 min at 4°C, cytosolic and organelle fractions were subjected to immunoprecipitation with antibodies to catalase and DHAPAT as described (*Tsukamoto et al., 1990*). $^{35}$S-labeled proteins were separated by SDS-PAGE and detected with an Autoimaging analyzer (Typhoon FLA-9500; GE Healthcare).

### Statistical analysis

Statistical analysis was performed using R software (http://www.r-project.org). Quantitative data were represented as means ± SEM from at least three independent experiments. Statistical significance was determined using a two-tailed unpaired Student's *t* test for comparisons between two groups or one-way ANOVA with Dunnett's post hoc test for more than two groups. P-values of <0.05 were considered statistically significant.

## Acknowledgements

We thank S Okuno and Y Nanri for technical assistance, the other members of Fujiki laboratory for expertise and discussions, and the members of Functional Cell Biology laboratory of Kyushu University for continuous supports. We thank Dr. Daniel Hess, Friedrich Miescher Institute for Biomedical Research, Basel, Switzerland, for LC/MS/MS analysis at the initial stage of this work. This work was supported in part by grants from the Ministry of Education, Culture, Sports, Science, and Technology of Japan; Grants-in-Aid for Scientific Research, MEXT KAKENHI Grant Number JP26116007 (to YF) and the Japan Society for the Promotion of Science Grants-in-aid for Scientific Research, Japan Society for the Promotion of Science KAKENHI Grants Numbers JP24770130, JP26440032, and JP17K07310 (to KO) and JP24247038, JP25112518, JP25116717, JP15K14511, JP15K21743, and JP17H03675 (to YF); grants from the Takeda Science Foundation (to YF), the Naito Foundation (to YF), the Japan Foundation for Applied Enzymology (to YF and KO), the Novartis Foundation (Japan) for the Promotion of Science (to YF), Joint Usage and Joint Research Programs of Institute of Advanced Medical Sciences, Tokushima University (to YF), and Qdai-jump Research Program in Kyushu University (to KO).

## Additional information

### Competing interests

Mahmoud El Shermely: Mahmoud El Shermely is affiliated with Basilea Pharmaceutica International Ltd. The author has no financial interests to declare. The other authors declare that no competing interests exist.

### Funding

| Funder | Grant reference number | Author |
|---|---|---|
| MEXT | JP26116007 | Yukio Fujiki |
| Japan Society for the Promotion of Science | JP24770130 | Kanji Okumoto |
| Japan Society for the Promotion of Science | JP26440032 | Kanji Okumoto |
| Japan Society for the Promotion of Science | JP17K07310 | Kanji Okumoto |
| Japan Society for the Promotion of Science | JP25112518 | Yukio Fujiki |
| Japan Society for the Promotion of Science | JP24247038 | Yukio Fujiki |
| Japan Society for the Promotion of Science | JP25116717 | Yukio Fujiki |
| Japan Society for the Promotion of Science | JP15K14511 | Yukio Fujiki |
| Japan Society for the Promotion of Science | JP15K21743 | Yukio Fujiki |
| Japan Society for the Promotion of Science | JP17H03675 | Yukio Fujiki |
| Takeda Science Foundation | | Yukio Fujiki |
| The Naito Foundation | | Yukio Fujiki |
| The Japan Foundation for Applied Enzymology | | Kanji Okumoto Yukio Fujiki |
| Novartis Foundation | | Yukio Fujiki |
| University of Tokushima | | Yukio Fujiki |
| Kyushu University | | Kanji Okumoto |

The funders had no role in study design, data collection and interpretation, or the decision to submit the work for publication.

### Author contributions

Kanji Okumoto, Conceptualization, Formal analysis, Funding acquisition, Validation, Investigation, Visualization, Methodology, Writing - original draft, Writing - review and editing; Mahmoud El Shermely, Conceptualization, Investigation, Methodology; Masanao Natsui, Toshihiro Marutani, Investigation; Hidetaka Kosako, Formal analysis, Validation, Investigation, Methodology, Writing - original draft, Writing - review and editing; Ryuichi Natsuyama, Formal analysis, Investigation; Yukio Fujiki, Conceptualization, Resources, Supervision, Funding acquisition, Validation, Methodology, Writing - original draft, Writing - review and editing

### Author ORCIDs

Kanji Okumoto (iD) https://orcid.org/0000-0002-0137-1431
Yukio Fujiki (iD) https://orcid.org/0000-0002-8138-6376

Decision letter and Author response
Decision letter https://doi.org/10.7554/eLife.55896.sa1
Author response https://doi.org/10.7554/eLife.55896.sa2

## Additional files

### Supplementary files
• Transparent reporting form

### Data availability

All data generated or analyzed during this study are included in the manuscript and supporting files. Source data files have been provided for Figures 2, 3, 4, and 6.

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
