## [Decision Letter]

**Acceptance summary:**

This study demonstrates that peroxisomal import is regulated by phosphorylation of the import machinery. The authors identify peroxide-triggered phosphorylation sites on Pex14 and show that phosphorylation at one such site impairs catalase import. A series of in vitro experiments suggests that phosphorylation impacts the formation of a stable complex between the substrate and import machinery, although the molecular details remain to be elucidated in future work. The reduced import of catalase during peroxide treatment is important for maintaining viability during this type of stress. The non-imported catalase is proposed to be important for clearing hydrogen peroxide from the cytosol. The conceptual advance that import is a regulated process of physiological importance is a new contribution to this field and opens up additional directions for study.

**Decision letter after peer review:**

Thank you for submitting your article "Peroxisome counteracts oxidative stresses by suppressing catalase import via Pex14 phosphorylation" for consideration by *eLife*. Your article has been reviewed by three peer reviewers, one of whom is a member of our Board of Reviewing Editors, and the evaluation has been overseen by David Ron as the Senior Editor. The reviewers have opted to remain anonymous.

The reviewers have discussed the reviews with one another and the Reviewing Editor has drafted this decision to help you prepare a revised submission.

As the editors have judged that your manuscript is of interest, but as described below that additional experiments are required before it is published, we would like to draw your attention to changes in our revision policy that we have made in response to COVID-19 (https://elifesciences.org/articles/57162). First, because many researchers have temporarily lost access to the labs, we will give authors as much time as they need to submit revised manuscripts. We are also offering, if you choose, to post the manuscript to bioRxiv (if it is not already there) along with this decision letter and a formal designation that the manuscript is 'in revision at *eLife*'. Please let us know if you would like to pursue this option. (If your work is more suitable for medRxiv, you will need to post the preprint yourself, as the mechanisms for us to do so are still in development.)

Summary:

This study investigates whether peroxisomal import is regulated by phosphorylation. The authors initially identify peroxide-triggered phosphorylation on Pex14 using Phos-tag gels, then identify the sites, and use mutations to probe their importance. The key discovery is that S232 on Pex14 is phosphorylated (among other sites) in response to H_2_O_2_, that a phosphomimetic mutation of this site impairs catalase import, and that the reduced import of catalase during H_2_O_2_ treatment is important to maintaining viability during the stress. in vitro interaction studies suggest that the phosphomimetic Pex14 mutant can bind Pex5L but does not form a stable ternary complex with Pex5L and catalase. The other phosphorylation sites seem to have less of an effect but may affect other PTS1 import substrates. The primary conceptual advance here is that peroxisome import machinery can be regulated by phosphorylation to affect import of some, but not other substrates. The referees agreed that the conceptual advance of identifying a new regulatory aspect of peroxisomal import is appropriate for publication in *eLife*, but that the data are currently insufficiently complete to fully support the manuscript's claims.

Essential revisions:

1) The mechanism proposed by the authors for regulation of catalase import involves Pex14 phosphorylation. Yet it is Pex5 that recognizes catalase in the cytosol and is required for chaperoning catalase across the peroxisome membrane. Thus, to understand the mechanism of regulation, their crucial in vitro experiments examining substrate-Pex5-Pex14 interactions need to use the appropriate substrate-Pex5 complexes. Mammals have two Pex5's, a short Pex5 responsible for PTS1 import, and Pex5L, which binds to Pex7 and helps guide PTS2-containing proteins into the peroxisome. The authors do not provide any justification in the manuscript for why Pex5L was used in the in vitro binding experiments, and they do not provide any comparative experiments using the short Pex5. The authors must address this concern in order to justify the extrapolation of the in vitro experiments to the situation in cells.

2) The phospho-serine rich site identified by the authors is predicted to be a PEST sequence by bioinformatic searches using the sequence for rat Pex14. PEST sequences are typically found on short-lived proteins and act as a signal for turnover by the proteasome or calcium-dependent calpain proteases. In several instances, Figure 1B, Figure 2A, Figure 2D, etc. it appears that oxidative stress results in a reduction of Pex14, consistent with a hypothesis that this proline and serine rich site is functioning like a PEST sequence. In Figure 4F, phosphorylated Pex14 is detected in the cytosolic fraction, which the authors claim is non-specific. An alternative explanation is that Pex14 is being extracted from the peroxisome and turned over upon H_2_O_2_ treatment. The dynamics of Pex14 turnover and its contribution to peroxisome import dynamics is not explored by the authors but has important implications for their hypothesis. The authors should carefully consider the possibility that phosphorylation regulates Pex14 turnover, which impacts import dynamics. If the authors have data on the turnover of Pex14 and its mutants under different conditions, this would be important to include. At the very least, this alternative explanation for regulation should be discussed in a revised manuscript.

3) The microscopy experiments present in Figure 3 and Figure 4 are not very convincing and are incomplete. It is difficult to see catalase in the cytosol in the S-to-D mutants. The control images stained for SKL are not shown, confounding the analysis. Further, the localization of the Pex14 mutants, while appearing punctate in the images, was not confirmed by colocalization with another PMP. Finally, equal expression of the different mutants relative to wild type was not verified (e.g., by SDS-PAGE analysis of parallel transfections). To make the experiment more complete, control SKL images need to be presented, the subcellular localization of the Pex14 mutants verified by colocalization with another PMP, and equal expression of the mutants verified by either quantification of the microscopy or SDS-PAGE.

4) Loading controls for the experiment in Figure 4D are needed to make this fully interpretable. Quantification of EGFP-PTS1 and HA-catalase in Figure 5C would be helpful to a reader.

5) Figure 4F is not convincing because the differences claimed are not very easy to appreciate and the degree of reproducibility of the small effects is not clear. To be convincing, this experiment needs to be quantified from multiple replicates and should be accompanied by total samples to show the levels of the proteins in each sample before fractionation.

6) The anti-His blot in Figure 2D is of poor quality and cannot be interpreted with confidence. The Pex14 blot is clear but is complicated by co-expression and partial co-migration of endogenous and exogenous Pex14 species. This experiment would be improved by either improving the quality of the anti-His blot, or perhaps if the authors preformed a His-pulldown followed by blotting to selectively visualize the exogenous proteins. The other option is to perform the experiment in cells lacking endogenous Pex14. Regardless of the approach taken, the authors should improve the quality of this important figure.

7) The claimed role of Pex13 is not clear from the results in Figure 5A. This experiment can be improved if the authors perform IP with phospho-specific antibody to substantiate the claim that phosphorylation of Pex14 alters it complex formation with Pex13. Alternatively, an IP via Pex13 could also be performed and show that the pS232 is coming down with Pex13.

8) The conclusion that phosphorylation is ERK mediated has been shown with a single inhibitor and should be extended to show this more directly by checking Pex14 in ERK knockout or siRNA cells.

---

## [Author Response]

Summary:This study investigates whether peroxisomal import is regulated by phosphorylation. The authors initially identify peroxide-triggered phosphorylation on Pex14 using Phos-tag gels, then identify the sites, and use mutations to probe their importance. The key discovery is that S232 on Pex14 is phosphorylated (among other sites) in response to H_2_O_2_, that a phosphomimetic mutation of this site impairs catalase import, and that the reduced import of catalase during H_2_O_2_ treatment is important to maintaining viability during the stress. in vitro interaction studies suggest that the phosphomimetic Pex14 mutant can bind Pex5L but does not form a stable ternary complex with Pex5L and catalase. The other phosphorylation sites seem to have less of an effect but may affect other PTS1 import substrates. The primary conceptual advance here is that peroxisome import machinery can be regulated by phosphorylation to affect import of some, but not other substrates. The referees agreed that the conceptual advance of identifying a new regulatory aspect of peroxisomal import is appropriate for publication in eLife, but that the data are currently insufficiently complete to fully support the manuscript's claims.

To address the issues raised by the editors and reviewers, we performed additional experiments and provided the main revised points as follows:

a) In vitro binding assays using Pex5S to show the similar property with Pex5L in the formation of Pex14-Pex5 complex with catalase and PTS1 protein (newly added Figure 5—figure supplement 1B; the reply to the Essential revision 1).

b) Assessing the protein level of Pex14 upon H_2_O_2_-treatment, showing little effect of Pex14 phosphorylation in its turnover (revised Figure 1D, newly added Figure 3A, and Figure 1—figure supplement 1A; the reply to the Essential revision 2).

c) Immunostaining images of catalase to clearly show the cytosolic catalase (revised Figures 3B and 4B) and those of PTS1 and PMP70 to verify the peroxisomal import of PTS1 proteins and the co-localization of Pex14 variants in PMP70-positive peroxisomes (newly added Figure 3—figure supplement 1B, C and Figure 4—figure supplement 1A, B; the reply to the Essential revision 3).

d) Several quantitative analyses to verify the peroxisomal import of catalase in the pulse-chase experiment (newly added Figure 4G; the reply to the Essential revision 5) and the in vitro interaction of Pex14-Pex5 complex with catalase or PTS1 protein (revised Figure 5C and newly added Figure 5—figure supplement 1B).

e) A clear immunoblot image with anti-His antibody to assess the site-specific phosphorylation of Pex14 (revised Figure 2E, reply to the Essential revision 6).

f) Immunoprecipitation with antibodies to Pex13 and Pex14-p232 to verify the relationship between Pex14 and Pex13 upon H_2_O_2_-treatment (revised Figure 5A; the reply to the Essential revision 7).

g) ERK2 knockdown in Fao cells to address the involvement of ERK2 in the H_2_O_2_-induced phosphorylation of Pex14 (newly added Figure 2—figure supplement 2; the reply to the Essential revision 8).

Essential revisions:1) The mechanism proposed by the authors for regulation of catalase import involves Pex14 phosphorylation. Yet it is Pex5 that recognizes catalase in the cytosol and is required for chaperoning catalase across the peroxisome membrane. Thus, to understand the mechanism of regulation, their crucial in vitro experiments examining substrate-Pex5-Pex14 interactions need to use the appropriate substrate-Pex5 complexes. Mammals have two Pex5's, a short Pex5 responsible for PTS1 import, and Pex5L, which binds to Pex7 and helps guide PTS2-containing proteins into the peroxisome. The authors do not provide any justification in the manuscript for why Pex5L was used in the in vitro binding experiments, and they do not provide any comparative experiments using the short Pex5. The authors must address this concern in order to justify the extrapolation of the in vitro experiments to the situation in cells.

While Pex5L is essential for PTS2-protein import as pointed out, both Pex5S and Pex5L are equally functional in PTS1-protein import and share the comparable interactions with the cargo PTS1-proteins and Pex14 (Otera et al.,1998, Saidowsky et al., 2001, Otera et al., 2012). In the original manuscript, we mainly used Pex5L as a PTS1 receptor in in vitro binding assays (Figure 5, B and C) and also showed that Pex5S less efficiently forms a ternary complex with Pex14-S232D and catalase (Figure 5—figure supplement 1A) as in the case of Pex5L (Figure 5B). To further address this issue, we performed in vitro binding assays for interactions between a full series of Pex14 variants, Pex5S, and PTS1-proteins (newly added Figure 5—figure supplement 1B). Pex5S represented an interaction profile similar to Pex5L, demonstrating that Pex5S and Pex5L similarly mediate the complex formation of Pex14 with catalase and PTS1 protein accompanied by the phosphorylation-dependent regulation. These results further supported the situation in cells upon an oxidative stress.

We added Figure 5—figure supplement 1B and revised the text involving this issue with additional introduction of two isoforms of Pex5 in the Results section and figure legends.

2) The phospho-serine rich site identified by the authors is predicted to be a PEST sequence by bioinformatic searches using the sequence for rat Pex14. PEST sequences are typically found on short-lived proteins and act as a signal for turnover by the proteasome or calcium-dependent calpain proteases. In several instances, Figure 1B, Figure 2A, Figure 2D, etc. it appears that oxidative stress results in a reduction of Pex14, consistent with a hypothesis that this proline and serine rich site is functioning like a PEST sequence. In Figure 4F, phosphorylated Pex14 is detected in the cytosolic fraction, which the authors claim is non-specific. An alternative explanation is that Pex14 is being extracted from the peroxisome and turned over upon H_2_O_2_ treatment. The dynamics of Pex14 turnover and its contribution to peroxisome import dynamics is not explored by the authors but has important implications for their hypothesis. The authors should carefully consider the possibility that phosphorylation regulates Pex14 turnover, which impacts import dynamics. If the authors have data on the turnover of Pex14 and its mutants under different conditions, this would be important to include. At the very least, this alternative explanation for regulation should be discussed in a revised manuscript.

As the reviewers pointed out, a putative PEST sequence was identified in the C-terminal region of rat Pex14 (amino-acid residues at 259-278) by an algorithm, PESTfind. We addressed the protein level of Pex14 upon H_2_O_2_ treatment by re-analysis of the original Figure 1C using exactly the same samples as those used in the original figure. In contrast to ~70% of total Pex14 was phosphorylated with a peak at 1 hour post-H_2_O_2_ challenge, total Pex14 level remained nearly constant in SDS-PAGE for 2 hours and reduced by approximately 30% at 5 hours after the H_2_O_2_ treatment, where a cytosolic protein LDH indicated a similar pattern (revised Figure 1D, corresponding to the original Figure 1C). We further verified any effect of H_2_O_2_ treatment on Pex14 stability and found that the exposure to H_2_O_2_ for 5 hours significantly lowered the protein level of Pex14 and concomitantly decreased LDH to a similar extent, thereby showing the relatively unaltered protein level of Pex14 (newly added Figure 1—figure supplement 1A). As the cells in these experiments were lysed in a constant volume of the buffer, we interpreted these results to mean that significant decrease of Pex14 observed at 5 hours after the H_2_O_2_ treatment is most likely due to less number of the cells by H_2_O_2_ toxicity, not by a preferential degradation of phosphorylated Pex14. This is further supported by the findings that both phosphorylation-deficient and phosphomimetic mutants of Pex14 were similarly expressed as wild-type Pex14 even in H_2_O_2_-treated ZP161 cells for 2 hours (newly added Figure 3A, lanes 9-16). Taken together, we think that phosphorylation-dependent regulation of Pex14 turnover has little effect on peroxisomal protein import and that phosphorylated Pex14 upon H_2_O_2_ treatment is most likely de-phosphorylated by currently yet identified phosphatase(s). Thus, the putative PEST sequence does not appear to function in the Pex14 degradation.

Regarding the band detected in the cytosolic fraction in anti-phospho-Pex14 blot (Figure 4F), it is most likely a non-specific band; 1) no cytosolic band was discernible with anti-Pex14 antibody at the position corresponding to that observed in anti-phospho-Pex14 blot and 2) the anti-phospho-Pex14-positive band in the cytosolic fraction seemed to migrate with slightly lower mobility than those in the organelle fractions in the respective conditions. We do not exclude the possibility that phosphorylation is involved in Pex14 turnover and a part of the phosphorylated Pex14 becomes extractable from the organelle membrane upon H_2_O_2_ treatment. These issues would be clarified in the future.

We added Figure 1—figure supplement 1A and revised Figure 1D (corresponding to the original Figure 1C) and the text addressing this issue in the Results section, Discussion section and figure legends.

3) The microscopy experiments present in Figure 3 and Figure 4 are not very convincing and are incomplete. It is difficult to see catalase in the cytosol in the S-to-D mutants. The control images stained for SKL are not shown, confounding the analysis. Further, the localization of the Pex14 mutants, while appearing punctate in the images, was not confirmed by colocalization with another PMP. Finally, equal expression of the different mutants relative to wild type was not verified (e.g., by SDS-PAGE analysis of parallel transfections). To make the experiment more complete, control SKL images need to be presented, the subcellular localization of the Pex14 mutants verified by colocalization with another PMP, and equal expression of the mutants verified by either quantification of the microscopy or SDS-PAGE.

We revised immunostaining images of catalase in ZP161 cells transiently or stably expressing Pex14 variants (Figure 3B, corresponding to Figure 3A in the original manuscript and Figure 4B, respectively). These improved the images clearly showing cytosolic catalase in ZP161 cells expressing phosphomimetic S-to-D mutants of Pex14, in which Pex14 variants were equally expressed as assessed by immunoblot analysis (transient expression in newly added Figure 3A and stable expression in Figure 4A). We provided representative images of PTS1 proteins (Figure 3—figure supplement 1B) used for quantifying the activity of PTS1 import in Pex14 variants (Figure 3C). Furthermore, we performed co-immunostaining of Pex14 variants with a peroxisomal membrane protein PMP70 in ZP161 cells and found that all Pex14 variants tested were co-localized with PMP70 in punctate structures, peroxisomes, as the wild-type Pex14 (Figure 3—figure supplement 1C). In ZP161-stable cell lines, WT-6, SA-13, and SD-30, exclusive peroxisomal localization of PTS1 (Figure 4—figure supplement 1A) and co-localization of Pex14 variants with PMP70 (Figure 4—figure supplement 1B) were likewise observed as in CHO-K1 cells. Collectively, these results showed that both S-to-A and S-to-D mutations did not affect peroxisomal localization and a protein level of Pex14 variants, confirming the specific suppression of catalase import by the phosphomimetic Pex14 variant with S232D.

We revised the text regarding New Figure 3A, New Figure 3—figure supplement 1BC, and Figure 4—figure supplement 1AB in the Results section and figure legends.

4) Loading controls for the experiment in Figure 4D are needed to make this fully interpretable. Quantification of EGFP-PTS1 and HA-catalase in Figure 5C would be helpful to a reader.

In Figure 4D, LDH was used as a loading control as well as a cytosolic protein for adequate subcellular fractionation. As the reviewers suggested, the recoveries of EGFP-PTS1 and HA-catalase in Figure 5C were quantified and represented at the bottom of these blots. We revised the text in the figure legends.

5) Figure 4F is not convincing because the differences claimed are not very easy to appreciate and the degree of reproducibility of the small effects is not clear. To be convincing, this experiment needs to be quantified from multiple replicates and should be accompanied by total samples to show the levels of the proteins in each sample before fractionation.

We agree with the reviewers’ concern in regard to an apparently low effect of H_2_O_2_ in catalase import. We think several reasons for this situation.

1) Catalase was imported into peroxisomes with substantially less efficiency than typical PTS1 proteins in in vitro import system (Fujiki and Lazarow, 1985, Miura et al., 1992) and in cells (Koepke et al., 2007), which is owing to the atypical PTS1 (KANL) of catalase. Therefore, the increase of catalase in organelle fraction was lower during 1 hour-chase even in the absence of H_2_O_2_.

2) We found in other experiments that medium changes with methionine- and cysteine-free medium and following washing steps attenuated the phosphorylation level of Pex14 upon H_2_O_2_ treatment. This would lead to a lower suppression effect of H_2_O_2_ on catalase import in pulse-chase experiment, where the experimental condition was best optimized including the essential processes for pulse-labeling. In the revised manuscript, we quantified ^35^S-catalase and ^35^S-DHAPAT and showed that the increase of peroxisomal import of catalase in normal condition was repressed by the H_2_O_2_-treatment (newly added Figure 4G). Unfortunately, we have been unable to perform further experiments by the restriction of our radioisotope-research facility under the epidemic of COVID-19. We hope that you would understand our situation. We think that in addition to the pulse-chase experiments, other several results including the H_2_O_2_-induced phosphorylation of endogenous Pex14 and suppressive phenotypes of phosphomimetic Pex14 variants in catalase import convincingly support the regulated import of catalase in cells in a Pex14 phosphorylation-dependent manner.

We added Figure 4G and revised the text regarding this issue in the Results section and figure legends.

6) The anti-His blot in Figure 2D is of poor quality and cannot be interpreted with confidence. The Pex14 blot is clear but is complicated by co-expression and partial co-migration of endogenous and exogenous Pex14 species. This experiment would be improved by either improving the quality of the anti-His blot, or perhaps if the authors preformed a His-pulldown followed by blotting to selectively visualize the exogenous proteins. The other option is to perform the experiment in cells lacking endogenous Pex14. Regardless of the approach taken, the authors should improve the quality of this important figure.

We had a clear immunoblot image with anti-His antibody by re-analysis of exactly the same samples as those used in original manuscript. We revised the Figure 2D (re-numbered as Figure 2E in the revised manuscript) by replacing the older anti-His blot with new, clear one that shows site-specific phosphorylation of Pex14.

7) The claimed role of Pex13 is not clear from the results in Figure 5A. This experiment can be improved if the authors perform IP with phospho-specific antibody to substantiate the claim that phosphorylation of Pex14 alters it complex formation with Pex13. Alternatively, an IP via Pex13 could also be performed and show that the pS232 is coming down with Pex13.

As the reviewers suggested, we performed immunoprecipitation analyses with antibodies to Pex13 and Pex14-p232 (Figure 5A, lanes 5-12). In immunoprecipitation of Pex13, phosphorylated Pex14 was more efficiently recovered from H_2_O_2_-treated cells, while unmodified Pex14 was discernible at a similar level regardless of H_2_O_2_ treatment (Figure 5A, lanes 5-8; compare the recovery of phosphorylated (solid arrowhead) to the unmodified Pex14 (open arrowhead) in Phos-tag PAGE (top panel). In immunoprecipitation with anti-Pex14-pS232 antibody, Pex13 was co-immunoprecipitated with phosphorylated Pex14, where a substantial amount of unmodified Pex14 was also associated with phosphorylated Pex14 (Figure 5A, lanes 9-12). Together with the data of immunoprecipitation with anti-Pex13 antibody in the original manuscript (Figure 5A, lanes 1-4), we concluded that H_2_O_2_-dependent phosphorylation of Pex14 increases its complex formation with Pex13 in peroxisomal membrane.

We revised Figure 5A by adding the immunoprecipitation data with antibodies to Pex13 and Pex14-p232 and the text regarding this issue in the Results section, Discussion section, Materials and methods section and figure legends.

8) The conclusion that phosphorylation is ERK mediated has been shown with a single inhibitor and should be extended to show this more directly by checking Pex14 in ERK knockout or siRNA cells.

As the reviewers suggested, we carried out knockdown of ERK by siRNA transfection. As simultaneous knockdown of ERK1 and ERK2 severely affected cell growth, ERK2, a mainly expressed ERK in Fao cells (Figure 2F in the revised manuscript) was selected as a target. Transfection of *ERK2* siRNA reduced the protein level of ERK2 by ~80%, but showed no apparent effect on phosphorylation level of Pex14 induced by H_2_O_2_ treatment (newly added Figure 2—figure supplement 2). Upon H_2_O_2_ treatment, phosphorylation of ERK1 was instead induced in ERK2-depleted cells at the comparable level to those of ERK2 in control siRNA-transfected cells, suggesting the complementation of ERK2 depletion by ERK1. Collectively, ERK2-knockdown by siRNA transfection did not further support the possible role of ERK in H_2_O_2_-induced phosphorylation of Pex14 observed in the experiment using kinase inhibitors (Figure 2F).

We added the data of ERK2-knockdown experiment as Figure 2—figure supplement 2. As described in the Discussion section in the initially submitted manuscript, addressing the kinase that directly phosphorylates Pex14 and the upstream kinase cascade is the most important task in the future works. At present, we think that more studies are required to show the involvement of ERK kinases in the H_2_O_2_-induced Pex14 phosphorylation. We revised the text regarding this issue in the Results section, Materials and methods section and figure legends.